# Unpacking Failure Modes of Generative Policies: Runtime Monitoring of Consistency and Progress

Christopher Agia[1], Rohan Sinha[1], Jingyun Yang[1], Zi-ang Cao[1],
Rika Antonova[1], Marco Pavone[1,2], Jeannette Bohg[1]

[1]Stanford University, [2]NVIDIA Research

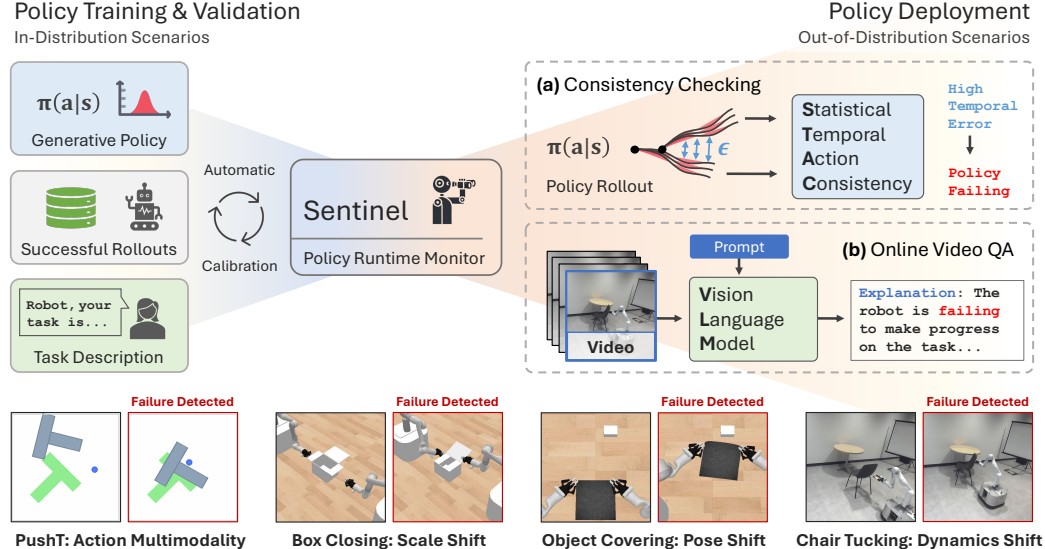

Figure 1: We present **Sentinel**, a runtime monitor that detects unknown failures of generative robot policies at deployment time. Constructing Sentinel requires only a set of successful policy rollouts and a description of the task, from which it detects diverse failures by monitoring **(a)** the temporal consistency of action-chunk distributions generated by the policy and **(b)** the task progress of the robot(s) through video QA with Vision Language Models.

**Abstract:** Robot behavior policies trained via imitation learning are prone to failure under conditions that deviate from their training data. Thus, algorithms that monitor learned policies at test time and provide early warnings of failure are necessary to facilitate scalable deployment. We propose **Sentinel**, a runtime monitoring framework that splits the detection of failures into two complementary categories: 1) Erratic failures, which we detect using statistical measures of temporal action consistency, and 2) task progression failures, where we use Vision Language Models (VLMs) to detect when the policy confidently and consistently takes actions that do not solve the task. Our approach has two key strengths. First, because learned policies exhibit diverse failure modes, combining complementary detectors leads to significantly higher accuracy at failure detection. Second, using a statistical temporal action consistency measure ensures that we quickly detect when multimodal, generative policies exhibit erratic behavior at negligible computational cost. In contrast, we only use VLMs to detect failure modes that are less time-sensitive. We demonstrate our approach in the context of diffusion policies trained on robotic mobile manipulation domains in both simulation and the real world. By unifying temporal consistency detection and VLM runtime monitoring, Sentinel detects 18% more failures than using either of the two detectors alone and significantly outperforms baselines, thus highlighting the importance of assigning specialized detectors to complementary categories of failure. **Qualitative results** are made available at sites.google.com/stanford.edu/sentinel.

**Keywords:** Failure Detection, Generative Policies, Vision Language Models

8th Conference on Robot Learning (CoRL 2024), Munich, Germany.

# 1 Introduction

Imitation learning represents one of the simplest yet most effective ways of learning robot control behaviors from data. Herein, generative modeling techniques have enabled robot policies to learn from highly heterogeneous, multimodal demonstration data collected by humans [1, 2, 3], showing early signs of generalization to novel environments [4] and embodiments [5]. When deploying robots beyond controlled lab settings, however, even the most powerful generative policies will eventually encounter *out-of-distribution* (OOD) test cases—scenarios that differ from the training data—on which their behavior becomes unpredictable [6]. In response, we will require methods that monitor the behavior of learned, generative polices at deployment time to detect whether they are failing as a result of distribution shift.

Identifying when a learned model behaves unreliably is typically framed as an OOD detection problem, for which a taxonomy of methods exist [7, 8]. While these methods can signal distribution shift [9, 10] or quantify uncertainty [11, 12] *w.r.t.* individual input-output samples, they do not fully characterize closed-loop policy failures that arise from multiple, time-correlated prediction errors along a trajectory rollout. Action multimodality further complicates the failure detection problem: that is, actions sampled from multimodal generative policies can vary greatly from one timestep to the next, leading to complex runtime behaviors and, by extension, diverse failure modes compared to previous model-free policies [13, 14]. Therefore, the special case of generative robot policies necessitates the design of new failure detectors suited to their multimodal characteristics and closed-loop operational nature in deployment.

In this work, we present **Sentinel**, a runtime monitoring framework that splits the task of detecting generative policy failures into two complementary categories. The first is the detection of failures in which the policy exhibits erratic behavior as characterized by its temporal inconsistency. For example, the robot may collide with its surroundings if the policy's action distributions contain conflicting action modes across time. To detect erratic failures, we propose to evaluate *how much* a generative policy's action distributions are changing across time using S̲tatistical measures of T̲emporal A̲ction C̲onsistency (STAC). The second category is the detection of failures in which the policy is temporally consistent but struggles to make progress on its task. For example, the robot can stall in place or drift astray if the policy produces constant outputs. We propose to detect task progression failures (undetectable by STAC) zero-shot with Vision Language Models (VLMs), which can distinguish off-nominal behavior when prompted to reason about the robot's progress in a video question answering setup. Notably, one would want to catch erratic failures (the first category) fast, whereas task progression failures (the second category) do not require immediate intervention.

Our contributions are three-fold: 1) A formulation of failure detection for generative policies that splits failures into two complementary categories, thus admitting the use of specialized detectors toward system-level performance increases (i.e., a *divide-and-conquer* strategy); 2) We propose STAC, a novel temporal consistency detector that tracks the statistical consistency of a generative policy's action distributions to detect erratic failures; 3) We propose the use of VLMs to monitor the task progress of a policy over the duration of its rollout, and we offer practical insights for their use as failure detectors. Provided with only a set of successful policy rollouts and a natural language description of the task, **Sentinel** (which runs STAC and the VLM monitor in parallel) detects over 97% of unknown failures exhibited by diffusion policies [1] across simulated and real-world robotic mobile manipulation domains.

# 2 Related Work

Advances in **robot imitation learning** include new policy architectures [1, 15, 16, 17], hardware innovations for data collection [18, 19, 20], community-wide efforts to scale robot learning datasets [3, 5, 21], and training high-capacity behavior policies on these datasets [2, 4, 22, 23]. Of recent interest is the use of generative models [24, 25, 26, 27] to effectively learn from heterogeneous and multimodal datasets of human demonstrations [1, 5, 2, 23]. *Generative policies* thereby learn to represent highly multimodal distributions from which diverse robot actions can be sampled. While state-of-the-art generative policies demonstrate remarkable performance, their inherent multimodality results in more stochastic runtime behavior than that of previous model-free policies [13, 14, 28, 29, 30, 31]. In this work, we focus on characterizing the behavior of generative robot policies for failure detection.

Despite recent progress, it is well known that **learned policies may fail** beyond their training distribution [6, 7, 8], in part due to compounding prediction errors on states induced by the policy [32, 33]. As such, a recent work proposes to bound the performance of imitation learned policies prior to deployment [34]. Other works propose to retrain the policy on OOD states using corrective supervision from humans [35, 36, 37, 38, 39]. Notably, these methods apply *after* failures have occurred, maintaining the need for runtime monitors that detect policy failures and prevent their downstream consequences. Thus, our focus can be viewed as complementary to methods that learn post hoc from corrective feedback.

The existing literature on **out-of-distribution detectors** and **runtime monitoring** for learned models is highly diverse, spanning multiple categories of methods. Model-based methods (e.g., [40, 41]) are not directly applicable to the model-free policies we consider. Some methods only pursue failure modes that are known *a priori* [42, 43, 44, 45], whereas we seek to detect unknown failures at deployment time. Many OOD detection works detect dissimilarity from training data via reconstruction [9, 46] or embedding similarity [47, 10], however, observational differences may not always result in policy failure. Other methods directly quantify epistemic uncertainty [11, 12, 48, 49], but come with considerable computational expense or may not be suitable for autoregressive, generative policy architectures, e.g., diffusion policies [1]. Several works monitor symbolic states to detect manipulation failures [50, 51], but assume access to multiple sensor modalities (e.g., vision, haptic, proprioception) for symbolic state estimation. Most related to our approach are algorithms that perform consistency checks across sensor modalities [52] and time [53]. Different from these, we directly monitor the consistency of a learned policy's action distributions and its task progress to detect closed-loop failure.

There is a growing interest in the use of **Foundation Models** [54] toward increasing robustness in robotic systems. Large Language Models are used to detect anomalies [55, 56] and to replan under execution failures [56, 57, 58, 59]. Reward models in the form of visual representations [60, 61] or VLMs [62] could be repurposed for failure detection by thresholding predicted rewards. However, [62] shows that additional fine-tuning is required to obtain reliable reward estimates. Du et al. [63] fine-tunes a VLM for episode-level success classification using a human annotated dataset on the order of $10^5$ trajectories. In contrast to this work, we 1) focus on zero-shot assessment with VLMs, 2) seek to detect policy failure amidst task execution, and 3) consider the system-level role of VLMs operating alongside policy-level failure detectors, and as such, assign each detector to a specified category of failure.

## 3 Problem Setup

**Failure Detection**    The goal of this work is to detect when a generative robot policy $\pi(a|s)$ fails to complete its task. The policy operates within a Markov Decision Process (MDP) with a finite horizon $H$, but it may terminate upon completion of the task at an earlier timestep. Given an initial state $s_0$ representative of a new test scenario, executing the policy for $t$ timesteps produces a trajectory $\tau_t = (s_0, a_0, ..., s_t)$. We define **failure detection** as the task of detecting whether a trajectory rollout $\tau_H$ constitutes a failure at the earliest possible timestep $t$. To do so, we aim to construct a failure detector $f(\tau_t) \rightarrow \{\texttt{ok}, \texttt{failure}\}$ that, at each timestep $t$, can provide a classification as to whether the policy *will fail* if it continues executing for the remaining $H - t$ timesteps of the MDP. Note that the failure detector makes its assessment solely based on the history of observed states and sampled actions up to the current timestep $t$.

The failure detector may contain parameters that require calibration, such as a detection threshold (as in §4.1). Therefore, we assume a scenario in which the policy $\pi$ is first trained, then validated on test cases where it is expected to perform reliably. This validation process yields a small dataset of $M$ successful trajectories $\mathcal{D}_\tau = \{\tau^i\}_{i=1}^M$ that can be used to calibrate the failure detector $f$ (if it contains parameters). Intuitively, the dataset $\mathcal{D}_\tau$ characterizes nominal policy behavior within or near the distribution of states it has been trained on, which helps to ground the assessment of potentially OOD trajectories at test time.

We measure failure detection performance in terms of true positive rate (TPR), true negative rate (TNR), and detection time (DT). We count a true positive if the failure detector raises a warning at any timestep in a trajectory where the policy fails, the earliest of which counts as the detection time. We count a true negative if the failure detector never raises a warning in a trajectory where the policy succeeds. We refer to §B.4 for supporting definitions of policy failure and key performance metrics.

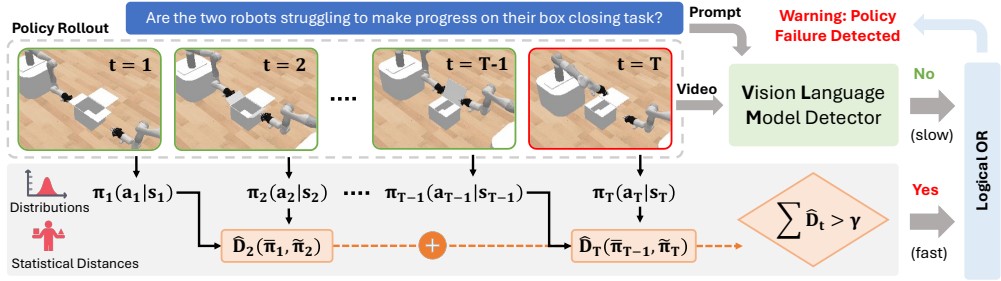

**Figure 3: Overview of Sentinel.** The images depict a policy rollout for timesteps $t = 1, ..., T$. Temporal Consistency Detector: At each timestep $t$, the state $s_t$ is passed to the generative policy to obtain action distributions $\pi_t$ between which statistical distances $\hat{D}_t$ are computed to measure temporal consistency. The statistical distances are summed up to the current timestep $T$ (as in Eq. 1) and thresholded by $\gamma$ to detect policy failure. Vision Language Model (VLM) Detector: The VLM classifies whether the policy is failing to make progress on its task given a video up to timestep $T$ and a description of the task. Execution stops if either detector raises a warning.

**Policy Formulation** We consider the setting where the policy $\pi$ is stochastic and predicts a sequence of actions (also referred to as an *action chunk* [18]) for the next $h$ timesteps. That is, the action sequence sampled at the $t$-th timestep, $a_t \sim \pi(\cdot|s_t)$, consists of $h$ actions $a_t := (a_{t|t}, a_{t+1|t}, ..., a_{t+h-1|t})$, where the notation $a_{t+i|t}$ denotes the action prediction for time $t+i$ generated at timestep $t$ (as in [64]). The actions $a_{\cdot|t} \in \mathcal{A}$ may correspond to e.g., end-effector poses or velocities. To control the robot, we sample an action sequence and execute the first $k < h$ actions, $a_{t:t+k|t}$, after which we re-evaluate the policy at timestep $t+k$. We visualize this receding horizon rollout for $k = 1$ in Fig. 2. Notably, $a_t$ and $a_{t+k}$ contain actions that temporally overlap for $h-k$ timesteps (i.e., at $a_{t+k:t+h-1|t}$ and $a_{t+k:t+h-1|t+k}$).

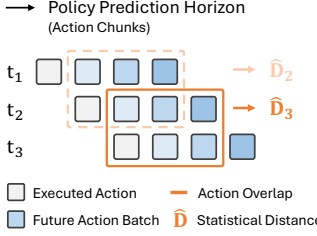

Figure 2: Action sequence prediction overlap during policy rollout.

Several recently proposed policy architectures achieve state-of-the-art performance by sampling action sequences using generative models [1, 18, 2], to which our approach is generically applicable. However, in this paper, we specify the failure detection problem for diffusion policies (DP) [1], which a) are stable to train and b) address action multimodality by representing the policy distribution with a denoising diffusion probabilistic model (DDPM) [26]. We note that the computationally intensive, iterative nature of the denoising process makes it challenging to directly apply existing OOD detection methodologies (e.g., [65]) to diffusion policies for failure detection, thus motivating several of our design decisions. Further details on the training and properties of these models are provided in §B.2.

## 4 Proposed Approach: Sentinel

The failure behavior of a generative policy by OOD conditions can be highly diverse, and we therefore argue that the desiderata for a failure detector may vary between qualitative types of failures. In this work, we propose to split the failure detection task into two complementary failure categories.

The first is the detection of failures resulting from erratic policy behavior, which may cause a robot to end up in states that are difficult or costly to reset from, knock over objects, or lead to safety hazards. Therefore, it is important to detect erratic behavior as quickly as possible (§4.1). The second category is the detection of failures in which the policy struggles to make progress on its task (hereafter referred to as *task progression failures*) but does so in a temporally consistent manner. For example, the policy may confidently place an object in the wrong location. Here, we must observe the robot over a longer period of time to identify that the policy is not making progress towards task completion (§4.2).

The **key insight** of our approach is that by defining one failure category as the complement of the other, it becomes trivial to combine failure detectors to form an accurate overall detection pipeline whilst satisfying the requirements of each failure category. Our full pipeline, **Sentinel**, is shown in Fig. 3.

## 4.1 STAC: Detecting Erratic Failures with Temporal Consistency

When a policy operates in nominal, in-distribution settings, it should complete its task in a temporally consistent manner. For example, a policy may plan to avoid an obstacle on the right or on the left, but not jitter between the two options. Moreover, as noted in [1], training a diffusion policy that predicts action sequences rather than individual actions encourages temporal consistency between action predictions.

Therefore, we propose to construct a quantifiable measure of temporal action consistency to detect whether the policy is behaving erratically, and hence, is likely to fail at the task. However, the multimodal distributional nature of DPs makes it difficult to directly compare two sampled actions $a_t \sim \pi(a_t|s_t)$ and $a_{t+k} \sim \pi(a_{t+k}|s_{t+k})$, e.g., throughout execution. This is because the actions may differ substantially along their prediction horizon when the policy commits or switches between different action modes, or simply due to randomness in sampling. Instead, we quantify erratic policy behavior with statistical measures of temporal action consistency (**STAC**, which we term our approach).

Let $\bar{\pi}_t := \pi(a_{t+k:t+h-1|t}|s_t)$ and $\tilde{\pi}_{t+k} := \pi(a_{t+k:t+h-1|t+k}|s_{t+k})$ be the marginal action distributions of the temporally overlapping actions between timesteps $t$ and $t+k$. We compute the temporal consistency between two contiguous timesteps $t$ and $t+k$ as $\hat{D}(\bar{\pi}_t, \tilde{\pi}_{t+k}) \geq 0$, where $\hat{D}$ denotes the chosen statistical distance function (e.g., maximum mean discrepancy, KL-divergence)[1]. In addition, we propose to take the cumulative sum of statistical distances along a trajectory as a measure of the overall temporal consistency in a policy rollout. At each policy-inference timestep $t = jk$ with $j \in \{0,1,...\}$, we compute the temporal consistency score as

$$\eta_t := \sum_{i=0}^{j-1} \hat{D}(\bar{\pi}_{ik}, \tilde{\pi}_{(i+1)k}). \tag{1}$$

Computing the consistency score in a cumulative manner has two advantages over thresholding the distance at each timestep individually. Firstly, it allows us to detect cases where the temporal consistency metric $\hat{D}$ is marginally larger than usual throughout the episode (e.g., jitter). Secondly, it allows us to detect instances where the policy is temporally inconsistent more often than in nominal scenarios.

At runtime, we raise a failure warning at the moment that $\eta_t$ exceeds a failure detection threshold $\gamma$, which we calibrate offline using the validation dataset of successful trajectories $\mathcal{D}_\tau$. Here, we first compute the cumulative temporal consistency scores throughout the entirety of the lengths $H_i \leq H$ of trajectories in $\mathcal{D}_\tau$, yielding $\{\eta_{H_i}^i\}_{i=1}^M$. Then, we set the threshold $\gamma$ to the $1-\delta$ quantile of $\{\eta_{H_i}^i\}_{i=1}^M$, where $\delta \in (0,1)$ is a hyperparameter. Intuitively, this ensures that the false positive rate (FPR)—the probability that we raise a false alarm and terminate on any trajectory that is i.i.d. with respect to $\mathcal{D}_\tau$—remains low, such that any warnings are likely failures. We can formalize this intuition using recent results from conformal prediction [66, 67]:

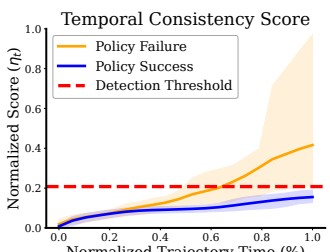

Figure 4: Temporal consistency scores grow faster when the policy fails. Error bars indicate the 5-th and 95-th score quantiles.

**Proposition 1** (STAC has low FPR). *Let $P_\tau$ denote the distribution of success trajectories in the validation dataset $\mathcal{D}_\tau = \{\tau^i\}_{i=1}^M \overset{iid}{\sim} P_\tau$. Then, the FPR—the probability of raising a false alarm at any point during an i.i.d. test trajectory $\tau \sim P_\tau$ of length $H' \leq H$—is at most $\delta$:*

$$\mathrm{FPR} := \mathbb{P}_{P_\tau}\big(\exists\, 0 \leq t \leq H' \text{ s.t. } \eta_t > \gamma\big) \leq \delta.$$

We refer to §A.1 for additional details on STAC and §D for a full statement and proof of Proposition 1.

## 4.2 Detecting Task Progression Failures with VLMs

A policy operating in out-of-distribution settings may not always fail by exhibiting erratic behavior that we can detect with STAC (§4.1). For example, suppose the policy confidently commits to the wrong

---

[1]Due to the iterative denoising procedure of the diffusion policy, analytically computing a distance $D(\bar{\pi}_t, \tilde{\pi}_{t+k})$ is challenging, as evaluating the densities of $\bar{\pi}_t$ and $\tilde{\pi}_{t+k}$ requires marginalizing out the intermediate diffusion steps and the non-overlapping actions. Instead, we approximate $D$ with its empirical counterpart $\hat{D}$ by sampling a batch of action sequences (parallelized on a GPU) at each timestep $t$ and $t+k$ rather than a single action sequence.

plan or produces approximately constant outputs. Detecting such failures requires an understanding as to whether or not the policy is progressing on its task, which necessitates a more comprehensive analysis of the robot's behavior within the context of its task specification. Therefore, we propose to use VLMs to monitor the task progress of the policy by providing them with the robot's image observations up to the current timestep as a video. We do so because recent work has shown that high-capacity VLMs possess robotics relevant knowledge and contextual reasoning abilities [68, 69, 63, 70, 62].

We formulate the detection of task progression failures as a chain-of-thought (CoT) [71], video question answering (QA) task [72, 73], reflecting current best practices in prompting. To capture a notion of *task progress*, the VLM must reason across time and in the context of the policy's task. Thus, we construct a prompt that contains a description of the task and the VLM's role as a runtime monitor. We query the VLM online using the text prompt and the history of observed images (i.e., a video) $I_{0:t} := (I_0, I_{\nu k}, I_{2\nu k}, ..., I_t)$ up to the current timestep $t$, where $\nu$ determines the frequency of the images relative to the execution horizon $k$ of the DP (§3). Differentiating between partial progress and task failure can be ambiguous for a slow moving robot, and thus, we also specify the current elapsed time $t$ and the time limit for the task $H$. This enables the VLM to gauge whether the rate at which the robot is executing will result in a timely task completion. After performing a CoT analysis, the VLM concludes with a classification in $\{$ok, failure$\}$. For additional details on the VLM and prompt, please see §A.2.

At the time of writing, cloud-querying a state-of-the-art VLM for video QA incurs significant latency (e.g., GPT-4o's mean response time was $14.0$s). However, we emphasize that VLM inference latency is a lesser concern for detecting task progression failures because they are likely to occur at longer timescales and do not require urgent intervention. In contrast, we assign the rapid detection of erratic failures to STAC (§4.1). Notably, the fast and slow detection requirements of our complementary failure categories mean that STAC and the VLM can operate at different timescales, offering potential benefits such as reduced costs and a lower likelihood of false positives when they run in parallel (Fig. 3).

## 5 Experiments

We conduct a series of experiments to test our failure detection framework. These experiments take place in both simulation and the real world (Fig. 1), and host an extensive list of baselines. We refer to §B for a detailed description of our environments, hardware setup, baselines, and evaluation protocol.

**Environments.** We include the **PushT** domain from [1] to evaluate the detection of failures under action multimodality. The **Close Box** and **Cover Object** domains involve two mobile manipulators and thus present the challenge of a high-dimensional, 14 degree-of-freedom action space. We additionally conduct hardware experiments with a mobile manipulator for a nonprehensile **Push Chair** task. This task presents greater dynamic complexity than the simulation domains [74]. At test time, we generate OOD scenarios by randomizing a) the scale and dimensions of objects in **PushT** and **Close Box** and b) the pose of the object in **Cover Object** and **Push Chair** beyond the policy's demonstration data.

**Baselines.** We evaluate Sentinel (i.e., both STAC and the VLM) against baselines representative of multiple methodological categories in the OOD detection literature [7]. Intuitively, these categories represent different formulations of the failure detector's score function, responsible for computing the per-timestep scores that are then summed to compute the trajectory score as in Eq. 1. We consider score functions based on the embedding similarity of observed states *w.r.t.* $\mathcal{D}_\tau$ [47], the reconstruction error of actions sampled from the DP [75], and the output variance of the DP. To strengthen the comparison, we introduce a new baseline that uses the DDPM loss (Eq. 2) on re-noised actions sampled from the DP as the failure detector's score function. Where applicable, we implement temporal consistency variants of these baselines to ablate the design of STAC. Further details on these baselines are provided in §B.3.

**Evaluation Protocol.** We train a DP for each environment and use standard settings for the DP's prediction and execution horizon [1]. We use the same calibration and evaluation protocol across all failure detection methods. That is, we calibrate detection thresholds to the 95-th quantile of scores in a dataset $\mathcal{D}_\tau = \{\tau^i\}_{i=1}^M$ of $M = 50$ in-distribution rollouts for each simulated task and $M = 10$ in-distribution rollouts for the real-world task. Finally, we report standard detection metrics including TPR, TNR, Mean Detection Time, Accuracy, and Balanced Accuracy, following the definitions in §3.

| Category 1: Erratic Failures | Close Box: In-Distribution (Policy Success Rate: 91%) | | | Close Box: Out-of-Distribution (Policy Success Rate: 41%) | | | Close Box: Combined (Policy Success Rate: 67%) | | |
|---|---|---|---|---|---|---|---|---|---|
| **Failure Detector** | TPR ↑ | TNR ↑ | Det. Time (s) ↓ | TPR ↑ | TNR ↑ | Det. Time (s) ↓ | TPR ↑ | TNR ↑ | Accuracy ↑ |
| **Diffusion** Temporal Non-Distr. Min. | 1.00 | 0.97 | 5.00 | 1.00 | 0.27 | 12.35 | 1.00 | 0.77 | 0.85 |
| Diffusion Recon. [75] | 0.33 | 0.95 | 13.60 | 0.40 | 1.00 | 17.08 | 0.37 | 0.96 | 0.76 |
| Temporal Diffusion Recon. | 1.00 | 0.96 | 8.47 | 0.92 | 1.00 | 15.75 | 0.92 | 0.97 | 0.95 |
| DDPM Loss (Eq. 2) | 1.00 | 0.90 | 8.27 | 1.00 | 0.94 | 14.54 | 1.00 | 0.91 | 0.94 |
| Temporal DDPM Loss | 1.00 | 0.95 | 7.53 | 1.00 | 0.37 | 13.66 | 1.00 | 0.79 | 0.86 |
| Diffusion Output Variance | 0.33 | 0.94 | 14.00 | 0.28 | 1.00 | 17.27 | 0.26 | 0.96 | 0.72 |
| **Embed.** Policy Encoder | 0.25 | 0.98 | 16.27 | 1.00 | 0.00 | 1.59 | 0.94 | 0.70 | 0.78 |
| CLIP Pretrained | 1.00 | 0.95 | 15.73 | 1.00 | 0.00 | 8.20 | 1.00 | 0.68 | 0.79 |
| ResNet Pretrained | 1.00 | 0.95 | 17.87 | 1.00 | 0.00 | 15.51 | 1.00 | 0.68 | 0.79 |
| **STAC** STAC For. KL (Ours) | 1.00 | 0.90 | 6.60 | 0.99 | 0.85 | 14.04 | 0.99 | 0.89 | 0.92 |
| STAC Rev. KL (Ours) | 1.00 | 0.95 | 7.60 | 0.93 | 0.97 | 15.12 | 0.93 | 0.96 | 0.95 |
| **STAC MMD\* (Ours)** | 1.00 | 0.94 | 7.20 | 0.99 | 0.93 | 14.72 | 0.99 | 0.94 | **0.96** |
| **VLM** GPT-4o Image QA | 1.00 | 0.00 | 23.20 | 1.00 | 0.00 | 23.20 | 1.00 | 0.00 | 0.29 |
| **GPT-4o Video QA\* (Ours)** | 1.00 | 0.89 | 21.20 | 0.69 | 0.95 | 21.02 | 0.77 | 0.91 | 0.87 |
| **Sentinel (STAC MMD\* + GPT-4o Video QA\*)** | 1.00 | 0.86 | 5.47 | 1.00 | 0.90 | 14.25 | 1.00 | 0.87 | 0.91 |

Table 1: **Detecting erratic policy failures in the Close Box domain.** Results are averaged over 3 random seeds. Our temporal consistency detector, STAC, accounts for when a policy fails (high true positive rate) and when it generalizes to out-of-distribution test cases (high true negative rate), in contrast to embedding-based methods that associate state atypicality with policy failure (low true negative rate). Select baselines that accurately detect erratic policy failures in this domain experience a decrease in performance under multimodal conditions (i.e., PushT, as shown in Fig. 5), whereas STAC continues to exhibit strong performance across multiple domains. VLMs must reason over video to attain high true negative rates, as is necessary to combine them with STAC (see Fig. 6). Sentinel, which runs STAC and the VLM monitor in parallel, detects 100% of erratic policy failures in this domain.

# 6   Results

**STAC detects diffusion policy failures in multimodal domains.** Fig. 5 (Left) compares STAC against the best performing method of each baseline category in the **PushT** domain. Here, STAC is the only method to achieve a balanced accuracy of over 90%, indicating that temporal consistency (or lack thereof) is strongly correlated with success (or failure). Alternative output metrics, such as the DP's output variance, do not perform well because both successes and failures can exhibit high-variance outputs in multimodal

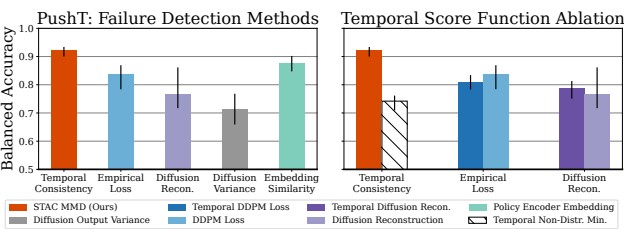

Figure 5: **Detecting failures in PushT.** Left: Our failure detector (STAC) which measures the temporal consistency of a generative policy outperforms several families of out-of-distribution detectors. Right: The best performance comes from measuring temporal consistency with statistical distance functions; augmenting baselines with temporal consistency does not always increase their performance.

domains. Interestingly, the embedding similarity approach performs strongly, which indicates that state dissimilarity *w.r.t.* the calibration dataset happens to be correlated with failure in this domain.

**Statistical measures of action similarity enable temporal consistency detection.** Fig. 5 (Right) ablates the design decisions of STAC. First, we observe that augmenting baselines with temporal consistency will at most marginally increase their performance. Second, using a non-statistical distance function (e.g., min. distance) to measure temporal action consistency performs worse than the baselines because it omits action multimodality. Thus, it is the combination of statistical distance functions with temporal consistency that yields the best result. We refer to §C.1 for an extended ablation of STAC.

**STAC accounts for OOD failures and generalization.** Results on the **Close Box** domains are shown in Table 1. STAC attains the highest accuracy in aggregate (96%). However, two of our newly proposed baselines—using the DDPM loss (Eq. 2) and a temporal reconstruction variant of [75]—also perform well, perhaps due to a decrease in action multimodality relative to **PushT**. Notably, we find that embedding similarity methods conflate OOD states with policy failure, resulting in false positives when the policy succeeds OOD. In contrast, STAC effectively differentiates OOD successes from failures.

**VLMs must reason across time.** In Table 1, we find that a state-of-the-art VLM (GPT-4o) struggles to identify task success when given only a single image. Instead, it must observe the robot over the extent of a policy rollout to more accurately reason about task progression and changes in state (resulting in a

91% TNR). While erratic failures are time-sensitive, they are visually more subtle and thus difficult for the VLM to detect (77% TPR). The robot takes more obviously wrong actions (e.g., stalling, drifting astray) in task progression failures (Fig. 6). As expected, the VLM has a significantly slower detection time relative to STAC, further highlighting STAC's value at quickly detecting erratic behavior.

**Sentinel: Combining STAC and VLMs for system-level performance increases.** We evaluate our failure detectors on distribution shifts that primarily lead to task progression failures in the **Cover Object** and **Close Box** domains. The result is shown in Fig. 6. STAC achieves a low TPR (44%) when the policy fails in a temporally consistent manner, whereas the VLM (`GPT-4o` for **Close Box**, `Claude` for **Cover Object**) accurately detects task progression failures. As a result, their combination (Sentinel) achieves a 93% TPR whilst incurring only a 7% increase in FPR. The rise in detection time indicates that both STAC (fast) and the VLM (slow) are contributing to the detection of failures.

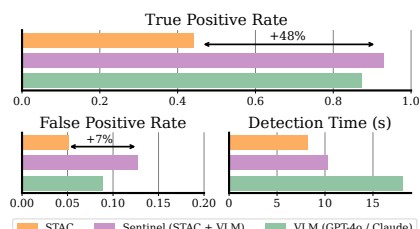

Figure 6: **Detecting task progression failures.** Combining VLMs with STAC yields an accurate overall detector (Sentinel) for both task progression and erratic failures (Table 1). See §C.2 for extended results and analysis.

**Sentinel detects real-world, generative policy failures.** We evaluate Sentinel on the **Push Chair** task across 10 successful and 10 failed policy rollouts. The results are shown in Table 2. When calibrated on only 10 successful in-distribution rollouts, STAC detects 80% of policy failures and raises only one false alarm (90% TNR). The VLM exhibits stronger performance in the real world (90% TPR, 100% TNR) than it does in the simulation domains, perhaps because real-world images constitute a lesser domain gap for visual reasoning compared to images rendered in simulation. Overall, Sentinel achieves a 95% detection accuracy, highlighting its efficacy for detecting failures in real-world robotic settings.

**Discussion.** Holistic analysis of Table 1, Fig. 6, and Table 2 shows that we can easily combine STAC and the VLM to yield a performant overall detector for both erratic and task progression failures, particularly because both detectors achieve a high overall TNR. Since all the baselines may 1) show low accuracy on either of the erratic failure domains (i.e., the multimodal **PushT** and **Close Box** domains) or 2) yield a low TNR, it is unclear how to combine them with other detectors in a way that outperforms Sentinel.

| Failure Detector | TPR ↑ | TNR ↑ | Det. Time (s) ↓ |
|---|---|---|---|
| Diffusion Output Variance | 0.60 | 0.90 | 10.67 |
| Temporal Non-Distr. Min. | 0.70 | 0.80 | 9.52 |
| **STAC Rev. KL** (Ours) | 0.80 | 0.90 | 9.83 |
| **GPT-4o Video QA** (Ours) | 0.90 | 1.00 | 12.89 |
| **Sentinel (STAC + GPT-4o)** | 1.00 | 0.90 | 9.60 |

Table 2: **Detecting real-world failures.** Sentinel demonstrates strong failure detection performance on the real-world Push Chair task, achieving an overall accuracy of 95%.

## 7    Conclusion

In this work, we investigate the problem of failure detection for generative robot policies. We propose Sentinel, a runtime monitor that splits the failure detection task into two categories: 1) Erratic failures, which we detect by measuring the statistical change of a policy's action distributions over time; 2) task progression failures, where we use Vision Language Models to assess whether the policy is consistently taking actions that do not solve the task. Our results highlight the importance of targeting complementary failure categories with specialized detectors. Future work includes the use of Sentinel to monitor high-capacity policies [2, 23], inform data collection, and accelerate policy learning.

**Limitations.** While categorizing erratic and task progression failures leads to accurate detection of failures across the domains we consider, these two failure categories may not be exhaustive. In the future, introducing additional categories or further partitioning existing ones might provide a broader coverage of failures, allow for more efficient failure detection, and inform mitigation strategies. Furthermore, our approach does not provide formal guarantees on detecting failures. However, providing such guarantees would require data of both successful and unsuccessful policy rollouts to calibrate the detector [67]. Although our detectors attain low false positive rates in aggregate, taking the union of their predictions may, in the worst case, increase the risk of false alarms. Thus, exploring more sophisticated ways to combine complementary failure detectors is a possible point of extension. Finally, our approach is not targeted at predicting failures before they occur, but instead focuses on detecting failures as they occur.

**Acknowledgments**

The authors would like to thank Rachel Luo and Apoorva Sharma for their early-stage feedback on the project. Toyota Research Institute, Toshiba, and the Stanford Institute for Human-Centered Artificial Intelligence provided funds to support this work. This work was also supported by Blue Origin and the National Aeronautics and Space Administration under the University Leadership Initiative program.

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

## Appendix Overview: Unpacking Failure Modes of Generative Policies

The appendix offers additional details with respect to the implementation of our failure detection framework (§A), the experiments conducted (§B), along with extended results and analysis (§C), and finally, supporting derivations (§D) for our proposed failure detectors. **Qualitative results** and a **video abstract** are made available at sites.google.com/stanford.edu/sentinel.

# A  Method Details: Sentinel

As shown in Fig. 3, the **Sentinel** runtime monitoring framework consists of the parallel operation of two complementary failure detectors, each assigned to the detection of a particular failure category of generative policies. The first is a temporal consistency detector that monitors for erratic policy behavior via statistical temporal action consistency (STAC) measures. The second is a Vision Language Model (VLM) that monitors for failure of the policy to make progress on its task. In this section, we provide additional details *w.r.t.* the implementation of STAC (§A.1) and the VLM runtime monitor (§A.2).

## A.1  Temporal Consistency Detection with STAC

**Background**    To summarize §3, STAC assumes the use of a stochastic policy $\pi$ that, at each policy-inference timestep $t$, predicts an action sequence for the next $h$ timesteps as $a_{t:t+h-1|t} \sim \pi(\cdot|s_t)$, executes the first $k$ actions $a_{t:t+k|t}$, before re-evaluating the policy at timestep $t+k$. Between two contiguous inference timesteps $t$ and $t+k$, sampled action sequences $a_{t+k:t+h-1|t}$ and $a_{t+k:t+h-1|t+k}$ (both in $\mathbb{R}^{(h-k)\times|\mathcal{A}|}$) overlap for $h-k$ timesteps. At a high-level, STAC seeks to quantify *how much* a generative policy's action distributions are changing over time. It does this by computing statistical distances between the distributions of overlapping actions, i.e., given $\bar{\pi}_t := \pi(a_{t+k:t+h-1|t}|s_t)$ and $\tilde{\pi}_{t+k} := \pi(a_{t+k:t+h-1|t+k}|s_{t+k})$, we compute $D(\bar{\pi}_t, \tilde{\pi}_{t+k})$.

**Hypothesis**    Our central hypothesis is that large statistical distances correlate with downstream policy failure. Intuitively, a predictive policy can be likened to possessing an internal world model that simulates how robot actions affect environment states. When the policy is in distribution, we expect this world model to be accurate, thus resulting in smaller statistical distances. More concretely, if the policy's internal model of state $s_{t+k}$ at timestep $t$ coincides with the actual observed state $s_{t+k}$ at timestep $t+k$, the distribution of actions $\tilde{\pi}_{t+k}$ should be well-represented by the distribution $\bar{\pi}_t$. As a result, the distance $D(\bar{\pi}_t, \tilde{\pi}_{t+k})$ will be small (for the right choice of statistical distance function $D$). Conversely, when the policy is out of distribution (OOD), its internal model of state $s_{t+k}$ at timestep $t$ may be inaccurate, yielding a divergence between $\bar{\pi}_t$ and $\tilde{\pi}_{t+k}$ and a larger statistical distance.

**Implementation Details**    As mentioned in §4.1, we propose to approximate $D(\bar{\pi}_t, \tilde{\pi}_{t+k})$ with an empirical distance function $\hat{D}$ instead of computing it analytically, as doing so presents the challenge of marginalizing out both the non-overlapping actions (between timesteps $t$ and $t+k$) and the intermediate steps of the diffusion process [76]. We found the following approximations to work well in practice:

- Maximum Mean Discrepancy (MMD) with radial basis function (RBF) kernels. We compute

$$\hat{D}(\bar{\pi}_t, \tilde{\pi}_{t+k}) = \mathbb{E}_{a_t, a'_t \sim \bar{\pi}_t}[k(a_t, a'_t)] + \mathbb{E}_{a_{t+k}, a'_{t+k} \sim \tilde{\pi}_{t+k}}[k(a_{t+k}, a'_{t+k})]$$

$$- 2\mathbb{E}_{a_t \sim \bar{\pi}_t, a_{t+k} \sim \tilde{\pi}_{t+k}}[k(a_t, a_{t+k})], \quad \text{where} \quad k(x, y; \beta_1) = \exp\left(-\frac{||x-y||^2}{\beta_1}\right).$$

  Here, $k : \mathbb{R}^{(h-k)\times|\mathcal{A}|} \times \mathbb{R}^{(h-k)\times|\mathcal{A}|} \to \mathbb{R}$ computes the similarity between two overlapping action sequences, and $\beta_1$ denotes the bandwidth of the RBF kernel. The expectations are taken over a batch of $B$ action sequences sampled from the generative policy.

- Forward KL-divergence via Kernel Density Estimation (KDE) of the policy distributions:

$$\hat{D}(\bar{\pi}_t, \tilde{\pi}_{t+k}) = \mathbb{E}_{a_{t+k} \sim \tilde{\pi}_{t+k}}\left[\log \frac{p(a_{t+k})}{q(a_{t+k})}\right],$$

  where $p$ and $q$ are KDEs of $\tilde{\pi}_{t+k}$ and $\bar{\pi}_t$ fit on a batch of $B$ action sequences sampled from each policy distribution, respectively. As before, we use Gaussian RBF kernels of the form $k(x, y; \beta_2)$, where $\beta_2$ denotes the bandwidth of the RBF kernels used for KDE.

- Reverse KL-divergence via KDE of the policy distributions:

$$\hat{D}(\bar{\pi}_t, \tilde{\pi}_{t+k}) = \mathbb{E}_{a_t \sim \bar{\pi}_t} \left[ \log \frac{p(a_t)}{q(a_t)} \right],$$

where $p$ and $q$ are KDEs of $\bar{\pi}_t$ and $\tilde{\pi}_{t+k}$ fit on a batch of $B$ action sequences sampled from each distribution, respectively, and all other parameters follow the forward KL definitions.

**Hyperparameters** The batch size $B$, MMD bandwidth $\beta_1$, and KDE bandwidth $\beta_2$ are hyperparameters that we select for a given environment. As expected, we found that larger batch sizes are necessary for accurate mean embeddings and density estimates in domains with higher degrees of multimodality (e.g., $B = 256$ action sequences for **PushT** and **Push Chair**). We also found that using either default settings or dynamic calibration techniques are sufficient to obtain suitable MMD and KDE bandwidth parameters $\beta_1$ and $\beta_2$, respectively. For example, setting $\beta_2$ in proportion to the maximum eigenvalue of the covariance of overlapping actions $a_{t+k:t+h-1|\cdot}$ sampled from $\bar{\pi}_t$ and $\tilde{\pi}_{t+k}$ worked well in multimodal domains. Further details on hyperparameters are provided in Table 3.

| Hyperparameters | PushT (↑ Multimodality) | Close Box & Cover Object (↓ Multimodality) | Push Chair (↑ Multimodality) |
|---|---|---|---|
| MMD + KDE batch size ($B$) | 256 | 32 | 256 |
| MMD bandwidth ($\beta_1$) | Median Heuristic [77, 78] | $1.0/\|\mathcal{A}\|$ | Median Heuristic |
| KDE bandwidth ($\beta_2$) | $\sqrt{\lambda_{\max}(\mathrm{Cov}(a_{t+k:t+h-1|\cdot}))}$ | 1.0 | $\sqrt{\lambda_{\max}(\mathrm{Cov}(a_{t+k:t+h-1|\cdot}))}$ |
| Policy action space ($\mathcal{A}$) | Linear Velocity | 2 x (Linear + Angular Velocity) | 1 x (Linear + Angular Velocity) |
| Policy prediction horizon ($h$) | 16 | 16 | 16 |
| Policy execution horizon ($k$) | 8 | 4 | 4 |

Table 3: **Hyperparameters settings** for temporal consistency detection with STAC.

**Additional Design Choices** There are several additional settings that one could adjust to increase STAC's detection performance on their task. First, filtering components of the policy's action space that are either noisy or discrete can increase the quality of the statistical distance score function. For example, the policy's action space in our robotic manipulator domains include end-effector linear and angular velocities, as well as a binary gripper command. However, when computing statistical distances, we omit all binary gripper commands. Next, reducing the execution horizon $k$ of the generative policy to compare action distributions that are closer in time can mitigate excessively large statistical distances in highly dynamic or stochastic environments. Likewise, comparing action distributions over a shorter prediction horizon $h$ may be suitable if the tails of predicted action sequences e.g., exhibit high variance.

## A.2 Runtime Monitoring with Vision Language Models

As described in §4.2, we formulate the detection of task progression failures as a chain-of-thought (CoT) [71], video question answering (Video QA) [73] task with VLMs. Below, we provide details on the implementation of our VLM runtime monitor and the prompt templates used in our experiments.

**Vision Language Models** In extended experiments (§C.2), we include variants of the VLM runtime monitor based on several models: OpenAI's GPT-4o, Anthropic's Claude 3.5 Sonnet, and Google's Gemini 1.5 Pro [79]. At the time of writing, these represent the current state-of-the-art VLMs for complex, multimodal reasoning tasks. We use consistent prompts across all models, however, we slightly vary the implementation of the monitor to reflect the suggested best practices of each VLM.

**Implementation Details** We propose to query the VLM online with a parsed text prompt describing the runtime monitoring task and the history of images (i.e., a video) $I_{0:t} := (I_0, I_{\nu k}, I_{2\nu k}, ..., I_t)$ captured by the robot's camera system up to the current timestep $t$. Here, the hyperparameter $\nu$ specifies the frequency of the images relative to the execution horizon $k$ of the generative policy (§3) for generality, as the video may be captured at a much higher frame rate than the policy's execution rate. In experiments, simply setting $\nu \in \{1, 2\}$ provided sufficient granularity for the VLM to identify the motion of the robot.

By making non-blocking API calls, the VLM runtime monitor can operate at relatively high frequencies. For example, the VLM can be queried at each policy-inference timestep $t = jk$ for $j \in \{0,1,...\}$ (i.e., at STAC's detection frequency) to provide a failure classification. However, depending on the task, doing so may neither be necessary nor desirable for two reasons. First, because task progression failures are likely to occur at longer timescales than erratic failures, querying the VLM at a reduced frequency might provide time for meaningful changes in state to occur. In turn, this would reduce redundancy in the VLM's predictions e.g., if no meaningful changes in state occurred since the last time it was queried. Second, while STAC—a statistical output monitor—can be evaluated at negligible cost (computationally and monetarily), querying state-of-the-art, closed-source VLMs may come with considerable expense. Since task progression failures are unlikely to require immediate intervention (in contrast to erratic failures), querying the VLM less often than STAC could be preferable. In experiments, we queried the VLM to detect task progression failures twice per episode.

The prompt template consists of three parts: 1) a brief description of the VLM's role as the runtime monitor of a robotic manipulator system, which it must execute by analyzing the attached video; 2) a description of the robot's task, the total amount of time that has elapsed[2], and time limit for the task (corresponding to the MDP horizon $H$ in §3); 3) instructions to elicit a CoT response, ensuring that the VLM describes and analyzes the motion of the robot and all task-relevant objects and outputs a classification that can be easily parsed. To remove ambiguity over the expected behavior of the robot and what constitutes task completion, we make sure that the task description is sufficiently detailed:

```
task_descriptions:
  cover: "hide the white box by covering it with the black blanket. The white box
        is located somewhere in front of the two robot arms and does not move. The
        black blanket starts directly in between the two robot arms"
  close: "close the white box by folding in the two smaller white side lids and
        the bigger white back lid. The white box is located in between the two
        robot arms and does not move. The robots should concurrently approach the
        side lids and push both side lids up, followed by approaching the back lid
        and folding up the back lid with both arms, without grasping the lids with
        the grippers"
  push_chair: "push the black chair into the circular table. The black chair
        starts directly in front of the robot. The robot should push black chair in
         a relatively straight line, without the chair rotating to the left or to
        the right, so that the seat of the chair is properly tucked under the
        circular table"
```

We elicit a four-step CoT response from the VLM that 1) generates a set of task-relevant questions whose correct answers would fully characterize the motion of the robot and all task-relevant objects in the video, 2) answers the task-relevant questions while providing fine-grained visual details, 3) analyzes the questions, answers, and elapsed time to identify whether the robot is making progress towards task completion within the episode time limit, and 4) concludes with an overall classification in {ok, failure}. Interestingly, we found that prompting the VLM to generate its own questions instead of manually specifying them leads to more accurate descriptions of the video and ensuing predictions.

**Prompt Template (Video QA)**

```
You are the runtime monitor for an autonomous mobile manipulator robot capable
    of solving common household tasks. A camera system captures a series of
    image frames (i.e., a video) of the robot executing its current task online.
     The image frames are captured at approximately 1Hz. As a runtime monitor,
    your job is to analyze the video and identify whether the robot is a) in
```

---

[2]Online runtime monitoring requires the VLM to differentiate whether a) the robot is still in progress of executing the task correctly (i.e., partial progress) or whether b) the robot will fail to complete the task (e.g., by stalling in a partially completed state). Differentiating between partial progress and task failure can be ambiguous for a slow moving robot, and thus, providing the VLM with the current elapsed time serves as a reference to gauge whether or not the rate at which the robot is executing the task will result in a timely task completion.

```
        progress of executing the task or b) failing to execute the task, for
        example, by acting incorrectly or unsafely.

The robot's current task is to {DESCRIPTION}. The robot may take up to {
    TIME_LIMIT} seconds to complete this task. The current elapsed time is {TIME
    } seconds.

Format your output in the following form:
[start of output]
Questions: First, generate a set of task-relevant questions that will enable you
     to understand the full, detailed motion of the robot and all task-relevant
     objects from the beginning to the end of the accompanying video.
Answers: Second, precisely answer the generated questions, providing fine-
    grained visual details that will help you accurately assess the current
    state of progress on the task.
Analysis: Assess whether the robot is clearly failing at the task. Since the
    video only represents the robot's progress up to the current timestep and
    the robot moves slowly, refrain from making a failure classification unless
    the robot is unlikely to complete the task in the allotted time. Explicitly
    note the amount of time that has passed in seconds and compare it with the
    time limit (e.g., x out of {TIME_LIMIT} seconds). Finally, based on the
    questions, answers, analysis, and elapsed time, decide whether the robot is
    in progress, or whether the robot will fail to complete its task in the
    remaining time (if any).
Overall assessment: {CHOICE: [ok, failure]}
[end of output]

Rules:
1. If you see phrases like {CHOICE: [choice1, choice2]}, it means you should
    replace the entire phrase with one of the choices listed. For example,
    replace the entire phrase '{CHOICE: [A, B]}' with 'B' when choosing option B.
     Do NOT enclose your choice in '{' '}' brackets. If you are not sure about
    the value, just use your best judgement.
2. Do NOT forget to conclude your analysis with an overall assessment. As
    indicated above with '{CHOICE: [ok, failure]}', your only options for the
    overall assessment are 'ok' or 'failure'.
3. Always start the output with [start of output] and end the output with [end of
     output].
```

### A.2.1   Prompt Ensembling

The Video QA failure detection task requires comprehensive and detailed reasoning of over potentially long sequences of images, which, at the time of writing, is a challenge for even the most capable VLMs. As such, we can expect the performance of the VLM runtime monitor to degrade when it is deployed in domains that are visually OOD *w.r.t.* the VLM's training data (e.g., images rendered in simulation or captured from unusual camera poses). In these settings, the VLM runtime monitor may provide a reasonable but imperfect signal on task progression failures, resulting in misclassifications.

To strengthen the reliability of our VLM runtime monitor, we propose a simple prompt ensembling strategy [80], whereby we construct multiple Video QA prompts, query the VLM with each prompt, and take the overall failure classification to be the *majority vote* of the predictions across all prompts. Intuitively, if the failure detectors associated with each individual prompt are fairly accurate to begin with, the resulting majority-vote detector will have an even higher probability of correctness.

In experiments, we only found it necessary to use prompt ensembling in the **Cover Object** domain. We construct two variants of the Video QA prompt (3 prompts total), each of which follow a similar CoT structure while including additional information to diversify the VLM's reasoning. The first prompt variant, **Video QA + Success Video**, includes a video of a successful policy rollout for the current task. This allows the VLM to distinguish off-nominal policy behavior at test time from nominal policy behavior illustrated in the example video. The second prompt variant, **Video QA + Goal Images**,

includes example images of the scene at the end of successful policy rollouts, which serve as a visual reference for task completion. In accordance with the assumptions of our framework, these prompt variants only require data associated with policy success to detect unknown failures at test time.

**Prompt Template (Video QA + Success Video)**

```
You are the runtime monitor... # Same as Video QA

To inform your analysis, you will be provided with an example video that shows
    the full motion of the robot and all task-relevant objects when the task is
    successfully executed. For example, the last image frame in the example
    video will show what the scene should look like at the end of a successsfully
     executed task. By comparing the current video with the example video, you
    may be able to visually distinguish when the robot is failing at the task
    versus when it is making steady progress or has completed.

The robot's current task is... # Same as Video QA

Questions: First, generate a set of task-relevant questions that will enable you
    to understand the full, detailed motion of the robot and all task-relevant
    objects from the beginning to the end of the accompanying video. In addition,
     generate questions that will enable you to identify any key similarities or
    differences between the current video and the example success video.
Answers: Second, precisely answer... # Same as Video QA
```

**Prompt Template (Video QA + Goal Images)**

```
You are the runtime monitor... # Same as Video QA

To inform your analysis, you will be provided with several example images that
    show what the scene (i.e., the robot and all task-relevant objects) should
    look like at the end of a successfully executed task. By comparing the last
    few image frames of the current video with these example images, you may be
    able to visually distinguish when the robot is failing at the task versus
    when it is making steady progress or has completed.

The robot's current task is... # Same as Video QA

Questions: First, generate a set of task-relevant questions that will enable you
    to understand the full, detailed motion of the robot and all task-relevant
    objects from the beginning to the end of the accompanying video. In addition,
     generate questions that will enable you to identify any key similarities or
    differences between the current video and the example images of
    successfully completed tasks.
Answers: Second, precisely answer... # Same as Video QA
```

# B  Experiment Details

## B.1  Environments

We provide additional details on the environments used to evaluate Sentinel. These environments vary in terms of their data distribution (e.g., multimodal or high-dimensional actions) and support different types of distribution shift (e.g., object scale, pose, dynamics), under which the behavior of generative diffusion policies can be methodically studied. The environments are visualized in Fig. 7.

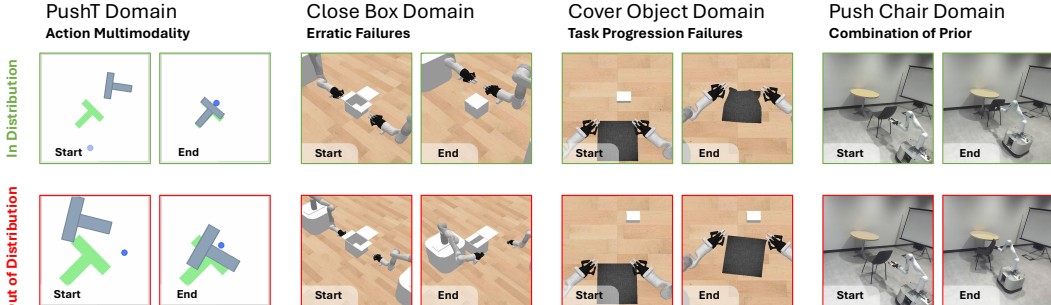

Figure 7: **Evaluation Domains.** We evaluate our failure detection framework across three simulation domains and one real-world domain. These domains provide coverage over different data distributions (e.g., action multimodality, high-dimensional actions) and modes of generative policy failure. For example, generative policies tend to fail erratically in the **Close Box** domain, but smoothly in the **Cover Object** domain. An effective failure detector should be performant across multiple domains, which entails coverage over heterogeneous failure modes.

### B.1.1 Simulation Domains

**PushT Domain** The policy is tasked with pushing a planar "T"-shaped object into a goal configuration. A trajectory is considered successful if the overlap between the "T"-shaped object and its goal exceeds 90% within 300 environment steps. The action space is the 2-DoF linear velocity of the end-effector. We generate OOD test scenarios by non-uniformly randomizing the scale and dimensions of the "T"-shaped object beyond the randomizations contained in the policy's demonstration data. The policy tends to fail by converging to a locally optimal configuration, where the "T" overlaps with its goal but in an incorrect orientation. Since the task can be solved in a number of ways, we include this domain to evaluate the performance of various score functions in the presence of action multimodality. We refer to [1] for the process of generating demonstration data in this domain.

**Close Box Domain** The policy is tasked with closing a box that has three lids. A trajectory is considered successful if all three lids are closed within 120 environment steps (24 seconds). The action space is the 14-DoF linear + angular velocities and gripper commands for the end-effectors of two mobile manipulators. Demonstration data is generated by an oracle policy that sets a series of waypoints for the end-effectors based on the initial state. We generate OOD test scenarios by non-uniformly randomizing the scale of the box beyond the randomizations contained in the policy's demonstration data. The policy tends to fail erratically when the robots e.g., collide with the box or its lids, however, task progression failures may also occur. This domain is primarily used to evaluate the detection of erratic policy failures on a bi-manual robotic system with a high-dimensional action space.

**Cover Object Domain** The policy is tasked with covering a rigid object with a cloth. A trajectory is considered successful if over 75% of the object is covered by the cloth within 64 environment steps (13 seconds). The action space and process of generating demonstration data is identical to that of **Close Box**. We generate OOD test scenarios by non-uniformly randomizing the position of the object beyond the randomizations contained in the policy's demonstration data. The policy tends to fail by releasing the cover before reaching the object. Hence, this domain is used to evaluate the detection of task progression failures, where reasoning over longer durations is required to assess task progress.

### B.1.2 Real-World Domains

**Mobile Robot Setup** We use a holonomic mobile base equipped with a Kinova Gen3 7-DoF arm. A single ZED 2 camera is fixed in the workspace to capture visual observations for the generative policy. The ZED 2 camera first generates a partial-view point cloud of the environment, from which we segment task-relevant objects using the Grounded Segment Anything Model [81] based on a natural

language description related to the task. The segmented point cloud serves as the visual input to the policy. Additionally, we use a motion capture system to track the pose of the mobile base. During evaluation, the policy processes the point clouds, predicts a sequence of 16 actions, of which the first 4 are executed on the robot. The mobile manipulator robot then maneuvers its arm according to these commands, adjusting the pose of the base if the end-effector moves outside a pre-defined workspace.

**Push Chair Domain**    The policy is tasked with tucking a chair into a table using a single-arm mobile manipulation platform. A trajectory is considered successful if the seat of the chair is properly tucked under the table by the end of the policy rollout. The action space is the 7-DoF linear + angular velocities and gripper command for the end-effector of the mobile manipulator robot. Demonstration data is extracted from human videos: we use an off-the-shelf hand detection model [82], an object segmentation model [83, 81], and a stereo-to-depth model to extract human hand poses and object point clouds from a subsampled set of frames in each of the 15 human demonstration videos. We generate OOD test scenarios by randomizing the initial pose of the chair beyond the randomizations contained in the demonstration data. The policy tends to fail erratically if the chair rotates away in either direction when pushed, but such failures are also visually apparent. Therefore, this task is used to test the efficacy of both STAC and the VLM runtime monitor in a dynamically complex [74], real-world setting.

## B.2  Diffusion Policies

We train a diffusion policy (DP) for each environment, using 200 demonstrations for the **PushT** domain, 50 demonstrations for each of the **Close Box** and **Cover Object** domains, and 15 demonstrations for the real-world **Push Chair** domain. In a DP, actions are generated by iteratively denoising an initially random action $a_t^N \sim \mathcal{N}(0,1)$ over $N$ steps as $a_t^N,...,a_t^0$, where $a_t^i$ with a superscript $i$ denotes the generated action sequence at the $i$-th denoising iteration. In an imitation learning setting, the DP's noise prediction network $\epsilon_\theta$ is trained to predict the random noise $\epsilon^i$ added to actions drawn from a dataset of expert demonstrations $\mathcal{D}_{\text{train}}$ by minimizing

$$\mathcal{L}_{\text{ddpm}} := \mathbb{E}_{(s,a^0)\sim\mathcal{D}_{\text{train}},\epsilon^i,i}\big[||\epsilon^i - \epsilon_\theta(\sqrt{\bar{\alpha}_i}a^0 + \sqrt{1-\bar{\alpha}_i}\epsilon^i,s,i)||^2\big], \tag{2}$$

where the constants $\bar{\alpha}_i$ depend on the chosen noise schedule of the diffusion process.

To increase the salience of distribution shift *w.r.t.* the position and scale of objects, we use point clouds as inputs to the policy instead of RGB images (i.e., a 5% increase in object scale may not be salient in an image). For simulation experiments, we use a diffusion policy architecture identical to the original paper [1] except for the visual encoder, where we substitute the ResNet-based encoder for a PointNet-based one: a 4-layer PointNet++ encoder [84] with hidden dimension 128. The output of this encoder is concatenated with the proprioceptive inputs and then fed to the noise prediction network. For real-world experiments, we use the recently proposed EquiBot diffusion policy architecture [85], which additionally incorporates SIM(3) equivariance into the diffusion process. We use EquiBot to evaluate our failure detectors on a current state-of-the-art approach for learning generative policies in the real world. All diffusion policies produce an action over $N = 100$ denoising iterations. Unless otherwise specified, we use standard settings for the prediction $h$ and execution horizon $k$ of the diffusion policy.

## B.3  Baselines

We outline the implementation details of our baselines as introduced in §5. First, with the exception of the VLM runtime monitors, all evaluated failure detection methods consist of computing a score $S(\cdot)$ at each policy-inference timestep in a rollout, taking the cumulative sum of scores up to the current timestep $t$, and then checking if the cumulative sum exceeds a calibrated threshold to detect policy failure. As such, the baselines differ in their *score function*, i.e., how they compute the per-timestep scores that are then summed and thresholded. Intuitively, a good score function should be well-correlated with policy failure, that is, it should output small values when the policy is succeeding and large ones when it is failing. For example, Fig. 4 demonstrates that STAC holds this property. We

baseline against an extensive suite of score functions, some of which we newly introduce for the case of generative diffusion policies, and others that are common in the OOD detection literature [7].

### B.3.1  STAC Baselines (Policy-Level Monitors)

- **Policy Encoder Embedding** quantifies the dissimilarity of the current point cloud observation $o_t$ *w.r.t.* to the point clouds in the calibration dataset of $M$ successful policy rollouts $\mathcal{D}_\tau = \{\tau^i\}_{i=1}^M$ (as described in §3) within the embedding space of the policy's encoder (here, $o_t$ denotes the point cloud input to the policy, which includes the point cloud at the current and previous timestep). More concretely, let $E$ be the policy's encoder, $z_t = E(o_t)$ be the current point cloud embedding, and $\mathcal{D}_z = E(\mathcal{D}_\tau)$ be the embeddings of all point clouds contained in the calibration dataset. We compute the per-timestep score as the Mahalanobis distance

$$S(z_t; \mathcal{D}_z) = \sqrt{(z_t - \mu_z)^T \Sigma_{zz}^{-1} (z_t - \mu_z)}, \tag{3}$$

where $\mu_z$ is the mean and $\Sigma_{zz}$ is the covariance of the embeddings in $\mathcal{D}_z$. At test time, we raise a failure warning if the cumulative score $\eta_t$ exceeds a calibrated detection threshold $\gamma$

$$\eta_t > \gamma, \quad \text{where} \quad \eta_t = \sum_{i=0}^{j-1} S(z_i; \mathcal{D}_z), \ t = jk.$$

Here, $\gamma$ is set to the $1 - \delta$ quantile of cumulative scores computed over the calibration trajectories $\{\eta_{|\tau^i|}^i\}_{i=1}^M$, where $\tau^i \in \mathcal{D}_\tau$. Importantly, when computing the calibration scores $\eta^i$, we do so in a *leave-trajectory-out* fashion: i.e., for a point cloud $o_t \in \tau^i$ where $\tau^i \in \mathcal{D}_\tau$, we compute the per-timestep score as $S(E(o_t); E(\mathcal{D}_\tau \setminus \tau^i))$. This ensures that the dissimilarity of observation $o_t$ is computed *w.r.t.* trajectories other than its own, which a) aligns with how scores are computed at test time and b) ensures that calibration scores are not trivially low.

  We experimented with alternatives to the Mahalanobis distance in Eq. 3, substituting it with top-$k$ scoring for $k \in \{1, 5, 10\}$ based on cosine similarity and L2 distance metrics. However, we found the Mahalanobis distance to be the most stable. We also evaluated variants of this baseline that compute the dissimilarity of the full policy state $s_t$ (including both the point cloud embedding and the robots' end-effector poses), but found equivalent performance.

- **CLIP Pretrained Embedding** quantifies the dissimilarity of the current image observation $I_t$ *w.r.t.* to the images in the calibration dataset $\mathcal{D}_\tau = \{\tau^i\}_{i=1}^M$ within the embedding space of a pretrained CLIP encoder [86]. The score function (Eq. 3) and calibration process are identical to those of **Policy Encoder Embedding**. Importantly, the encoder used here is trained with a representation learning objective, which results in a structured embedding space and more interpretable embedding similarity scores. In our experiments, we use the open-source `clip-vit-base-patch32` version of CLIP without any fine-tuning.

- **ResNet Pretrained Embedding** is identical to **CLIP Pretrained Embedding**, except it quantifies image-space dissimilarity using embeddings from a ResNet18 pretrained model [87].

- **Temporal Non-Distributional Minimum** is similar to STAC (§A.1) in that it seeks to compute a consistency score between overlapping actions $a_{t+k:t+h-1|t}$ and $a_{t+k:t+h-1|t+k}$ sampled from the generative policy at contiguous policy-inference timesteps $t$ and $t + k$, respectively. However, it does so by using a non-statistical distance function. In particular, this baseline computes the per-timestep temporal consistency score at timestep $t+k$ as

$$S(s_{t+k}) = \min_{b \in \{1..B\}} \left\| a_{t+k:t+h-1|t} - a_{t+k:t+h-1|t+k}^b \right\|,$$

where $a_{t+k:t+h-1|t+k}^b \sim \tilde{\pi}_{t+k}(\cdot | s_{t+k})$. That is, we sample a batch of $B$ action sequences at timestep $t + k$, compute their L2 distances *w.r.t.* the overlapping actions of the previously executed action sequence $a_{t+k:t+h-1|t}$, and return the L2 distance associated with the most similar action sequence. Intuitively, this baseline attempts to find the closest action sequence at timestep $t+k$ to the previously executed action sequence, while STAC attempts to quantify

how well the action distribution $\tilde{\pi}_{t+k}$ at timestep $t+k$ is represented in the distribution $\bar{\pi}_t$ at timestep $t$. The values of $B$ are provided in Table 3. The calibration and runtime procedures of this baseline are identical to those of STAC (§4.1).

- **Diffusion Reconstruction** adapts the diffusion-based OOD detection approach of Graham et al. [75] for the case of diffusion policies. Specifically, this baseline computes the reconstruction error on re-noised action sequences sampled from the diffusion policy as

$$S(s_t) = \mathbb{E}_{a^0 \sim \pi(\cdot|s_t), \epsilon^i, i}\left[\left\|a^0 - \epsilon_\theta^{i:0}(\sqrt{\bar{\alpha}_i}a^0 + \sqrt{1-\bar{\alpha}_i}\epsilon^i, s_t)\right\|^2\right], \qquad (4)$$

where $\epsilon_\theta^{i:0}$ denotes the reverse diffusion process from the $i$-th denoising iteration to the 0-th iteration, resulting in the reconstructed action. We approximate the expectation in Eq. 4 over a batch of $B = 256$ action sequences sampled from the diffusion policy, each re-noised for $i \in \{5, 10, 25, 50\}$ forward diffusion steps (also referred to as *reconstruction depths*). We experimented with several sets of reconstruction depths and found comparable performance. We note that this baseline comes with significant computational expense as it needs to perform the denoising process multiple times: i.e., if we would like to compute $R$ reconstructions, this baseline is approximately $R$ times more expensive than a single reverse diffusion process. The calibration and runtime procedures of this baseline are identical to those of STAC (§4.1).

- **Temporal Diffusion Reconstruction** is a temporal variant of **Diffusion Reconstruction** that also computes the reconstruction error on re-noised action sequences sampled from the diffusion policy, but reconstructs the action sequences conditioned on the previous state $s_t$ as

$$S(s_t, s_{t+k}) = \mathbb{E}_{a^0_{t+k:t+h-1|t+k} \sim \tilde{\pi}_{t+k}, \epsilon^i, i}\left[\left\|\hat{a}^0 - \epsilon_\theta^{i:0}(\sqrt{\bar{\alpha}_i}\hat{a}^0 + \sqrt{1-\bar{\alpha}_i}\epsilon^i, s_t)\right\|^2\right].$$

Here, $\hat{a}^0$ denotes the action sequence over which reconstructions are computed, concatenating the first $k$ (executed) actions sampled at timestep $t$ with following $h-k$ (predicted) actions sampled at timestep $t+k$: that is, $\hat{a}^0 = a_{t:t+k|t} \oplus a^0_{t+k:t+h-1|t+k}$. This step is necessary to ensure that the denoising process conditioned on $s_t$ only considers actions within the policy's prediction horizon. This baseline represents an alternative form of temporal consistency. Intuitively, it asks whether action sequences sampled at timestep $t+k$ would also be sampled at timestep $t$, to which the answer is likely yes if the policy is in distribution, and likely no if the policy is OOD—because the marginal distributions conditioned on $s_t$ versus $s_{t+k}$ may be different. The hyperparameters of this baseline follow those of **Diffusion Reconstruction**.

- **DDPM Loss** computes the empirical DDPM loss on re-noised action sequences sampled from the diffusion policy as

$$S(s_t) = \mathbb{E}_{a^0 \sim \pi(\cdot|s_t), \epsilon^i, i}\left[\left\|\epsilon^i - \epsilon_\theta(\sqrt{\bar{\alpha}_i}a^0 + \sqrt{1-\bar{\alpha}_i}\epsilon^i, s_t, i)\right\|^2\right].$$

Here, the expectation is taken over a batch of $B = 256$ sampled action sequences and 10 sampled denoising iterations $i \sim \mathcal{U}[0, N)$, where $N$ is the total number of denoising iterations (§B.2). We can think of this baseline as a more efficient version of **Diffusion Reconstruction**, since it directly quantifies the diffusion policy's performance on its training task without the need to reconstruct actions over numerous denoising iterations. The calibration and runtime procedures of this baseline are identical to those of STAC (§4.1).

- **Temporal DDPM Loss** is a temporal variant of **DDPM Loss** that also computes the empirical DDPM loss on re-noised action sequences sampled from the diffusion policy, but does so conditioned on the previous state $s_t$ as

$$S(s_t, s_{t+k}) = \mathbb{E}_{a^0_{t+k:t+h-1|t+k} \sim \tilde{\pi}_{t+k}, \epsilon^i, i}\left[\left\|\epsilon^i - \epsilon_\theta(\sqrt{\bar{\alpha}_i}\hat{a}^0 + \sqrt{1-\bar{\alpha}_i}\epsilon^i, s_t, i)\right\|^2\right],$$

where $\hat{a}^0 = a_{t:t+k|t} \oplus a^0_{t+k:t+h-1|t+k}$ (as defined in **Temporal Diffusion Reconstruction**). The hyperparameters of this baseline follow those of **DDPM Loss**, over which it is expected to offer advantages via temporal consistency.

- **Diffusion Output Variance** computes the variance over $B$ action sequences sampled from the diffusion policy and thresholds it *w.r.t.* the $1-\delta$ quantile of sample variances computed over the calibration dataset $\mathcal{D}_\tau$. This baseline reflects an alternative output metric to temporal consistency that can be monitored to detect policy failure. While computing output variances might bear resemblance to ensemble methods [12], we note that this approach does not quantify epistemic model uncertainty. Doing so would require training multiple diffusion policies and performing inference with each at test time, which we avoid due to computational expense. The hyperparameters of this baseline are identical to those of STAC (see Table 3).

**Discussion on Baselines**    First, we highlight that the embedding-based approaches predict failure solely based on the dissimilarity or atypicality of the current state. Hence, these baselines are not *policy aware*: they may raise failure warnings for states that are dissimilar from those contained in the calibration dataset $\mathcal{D}_\tau$ without understanding how the policy behaves in those states. In some cases, the policy may still succeed or generalize to minor distribution shifts in state, causing the detection performance of these baselines to significantly diminish. The reconstruction-based approaches may account for the generalization characteristics of the policy but come with computational expense, which may prohibit their use in real-time settings. The DDPM loss approaches present the next best alternative to STAC, as their score functions coincide with the diffusion policy's training task and can be computed at negligible computational cost. However, we note that the DDPM loss baseline is specific to diffusion policies, whereas STAC is agnostic to the generative policy formulation.

### B.3.2    VLM Baselines (Task-Level Monitors)

As described in §4.2, we propose to monitor the task progress of a generative policy by zero-shot prompting a VLM to analyze a video of the robot's execution up to the current timestep. We contrast the performance of our Video QA approach with a variation, Image QA, that queries the VLM using only $I_t$, the image recorded at the current timestep $t$, rather than the full video $I_{0:t}$ This baseline is used to evaluate the importance of video-based reasoning compared to single images. We construct the Image QA prompt by minimally modifying the Video QA prompt (§A.2) as shown below:

**Prompt Template (Image QA)**

```
You are the runtime monitor for an autonomous mobile manipulator robot capable
    of solving common household tasks. A camera system captures image frames (at
     approximately 1Hz) of the robot executing its current task online. As a
    runtime monitor, your job is to analyze the most recent image frame and
    identify whether the robot is a) in progress of executing the task or b)
    failing to execute the task, for example, by acting incorrectly or unsafely.

The robot's current task is to {DESCRIPTION}. The robot may take up to {
    TIME_LIMIT} seconds to complete this task. The current elapsed time is {TIME
    } seconds.

Format your output in the following form:
[start of output]
Questions: First, generate a set of task-relevant questions that will enable you
     to thoroughly analyze the image frame and identify what actions the robot
    has taken so far.
Answers: Second, precisely answer the generated questions, providing fine-
    grained visual details that will help you accurately assess the current
    state of progress on the task.
Analysis: Assess whether the robot is clearly failing at the task. Since the
    image frame only represents the robot's progress up to the current timestep
    and the robot moves slowly, refrain from making a failure classification
    unless the robot takes unsafe actions or is unlikely to complete the task in
    the allotted time. Explicitly note the amount of time that has passed in
    seconds and compare it with the time limit (e.g., x out of {TIME_LIMIT}
```

```
      seconds). Finally, based on the questions, answers, analysis, and elapsed
      time, decide whether the robot is in progress, or whether the robot will
      fail to complete its task in the remaining time (if any).
Overall assessment: {CHOICE: [ok, failure]}
[end of output]

Rules:
1. If you see phrases like {CHOICE: [choice1, choice2]}, it means you should
   replace the entire phrase with one of the choices listed. For example,
   replace the entire phrase '{CHOICE: [A, B]}' with 'B' when choosing option B.
    Do NOT enclose your choice in '{' '}' brackets. If you are not sure about
   the value, just use your best judgement.
2. Do NOT forget to conclude your analysis with an overall assessment. As
   indicated above with '{CHOICE: [ok, failure]}', your only options for the
   overall assessment are 'ok' or 'failure'.
3. Always start the output with [start of output] and end the output with [end of
   output].
```

## B.4   Evaluation Protocol

### B.4.1   Definition: Policy Failure

Consider a policy $\pi(a|s)$ that operates within a finite-horizon Markov Decision Process (MDP): a 5-tuple $\langle \mathcal{S}, \mathcal{A}, T, R, H \rangle$, where $\mathcal{S}$ and $\mathcal{A}$ are the state and action spaces, $T(s'|s,a)$ is the transition model, $R(s,a,s')$ is the reward model, and $H$ is the MDP horizon. Given an initial state $s_0$ representative of a new test scenario, executing the policy for $t$ timesteps produces a trajectory $\tau_t = (s_0, a_0, ..., s_t)$. The trajectory's *return* is defined as the cumulative sum of rewards: $R(\tau_t) = \sum_{t'=0}^{t-1} R(s_{t'}, a_{t'}, s_{t'+1})$.

We define **policy failure** simply in terms of task completion. More formally, given a defined success threshold $R_\tau$, the policy fails if the return on its trajectory $\tau_t$ does not exceed the success threshold within the MDP horizon: $R(\tau_t) < R_\tau$ where $t \geq H$. In the simplest case, the success threshold $R_\tau$ equals 1, and the reward model $R(s,a,s')$ equals 1 *iff* the task is complete at state $s'$. For example, if the robot is tasked with picking up a cup and receives a reward of 1 only once the cup is firmly grasped. In experiments, we adhere to this definition of policy failure and threshold trajectory returns as $R(\tau_H) < R_\tau$ to compute ground-truth labels for whether or not a policy failed in a trajectory $\tau_H$.

**Relation to failure detection:** In §3, we provide a definition of the failure detection task—to detect whether a trajectory $\tau_H$ constitutes a policy failure at the earliest possible timestep $t$—that is different from detecting the specific timestep at which (or before) the policy "fails." Doing so removes the need to manually specify task-specific failure criteria required to e.g., label each timestep in a trajectory. While the goal of our failure detectors is thus to flag failure episodes, it is still beneficial to catch failures at the earliest possible timestep, which is why a) our above definition of policy failure is formulated in terms of a partial trajectory $\tau_t$ up to the current timestep $t \leq H$ of the MDP, b) we propose an online detection scheme that monitors for failure at each timestep $t$ based on the trajectory up to the current timestep (i.e., $f(\tau_t) \rightarrow \{\texttt{ok}, \texttt{failure}\}$), and c) we report the detection time as a performance metric.

### B.4.2   Constructing the Calibration Dataset

Calibrating STAC and its baselines requires a small dataset of successful policy rollouts $\mathcal{D}_\tau = \{\tau^i\}_{i=1}^M$, which provide grounding on the nominal, in-distribution behavior of the policy. This allows us to evaluate the test-time behavior of a potentially failing policy *w.r.t.* its known nominal behavior.

**Calibration Data Quality**   We found it important to ensure the quality of trajectories $\tau^i \in \mathcal{D}_\tau$. Specifically, trajectories in which the policy succeeds but in an undesired or unacceptable manner should not be used for calibration. For example, the policy may solve the **Close Box** task (Fig. 7) but

damage the lids of the box in the process. Including such a trajectory in the calibration dataset would define this behavior as *nominal* and degrade the sensitivity of the detectors at test time. Returning to our example, the detectors may not raise a failure warning if the policy damages a box at test time.

**Collecting the Calibration Dataset**     In practice, such a calibration dataset could be collected during a policy validation phase prior to deployment. For instance, we collect $M = 50$ successful policy rollouts for each simulation domain, manually filtering episodes where the policy succeeded with unacceptable behavior (e.g., with jitter). We hypothesize that the performance of the detectors *w.r.t.* the number of rollouts $M$ is task specific. For example, a smaller calibration dataset may be sufficient for tasks with low variability (i.e., in a single, structured environment), while a larger dataset may be necessary if the policy is to be deployed at scale. We note, however, that increasing the calibration dataset size may be desirable to achieve stronger conformal guarantees on the detector's FPR (as derived in §D).

**Calibrating on Demonstration Data**     Finally, in attempt to eliminate the need to collect an additional calibration dataset of successful policy rollouts, we experimented with variants of STAC that directly calibrated on trajectories contained in the policy's demonstration dataset. However, doing so led to a significant increase in the detector's FPR. We attribute this to the well-known covariate shift problem for imitation learned policies [32, 33]. That is, their prediction error increases quadratically on states induced under the policy, causing the detectors' to mistake successful test-time rollouts for failures.

### B.4.3   Testing & Evaluation

Instead of evaluating the failure detectors online (i.e., during policy rollouts), we collect several test datasets of policy rollouts, which consist of both successes and failures. Each trajectory is labeled either success or failure by thresholding the return at the final state of the episode (as detailed in §B.4.1). We then perform offline evaluation of the failure detectors by invoking them at each timestep of the trajectory, which allows us to identify the first timestep at which the detectors issue a warning.

### B.4.4   Reported Metrics

We expand on the metrics outlined in §3. We first define a *positive* as a trajectory where the policy fails and a *negative* as a trajectory where the policy succeeds. A true positive is counted if the failure detector raises a warning at any timestep in a trajectory where the policy fails. A true negative is counted if the failure detector never raises a warning in a trajectory where the policy succeeds. The definitions for false positive and false negative follow accordingly. Detection time is defined as the earliest timestep in which the failure detector raises a warning in a trajectory where the policy fails.

In our experiments, we report true positive rate (TPR), true negative rate (TNR), false positive rate (FPR), detection time (DT), accuracy, and balanced accuracy. TPR measures the number of true positives (detected failures) over total number of positives (failures). TNR measures the number of true negatives (detected successes) over total number of negatives (successes). FPR measures the number of false positives (false alarms) over the total number of negatives (successes). Accuracy and balanced accuracy account for both the TPR and TNR of the detector. However, we report balanced accuracy when the test set contains a non-negligible imbalance of positive and negative trajectories.

## C   Additional Results

### C.1   Ablation Experiments on STAC

**Does STAC's performance depend on the policy's prediction and execution horizon?**

We conduct an ablation study on the **PushT** domain to test how the performance of STAC varies with respect to the prediction horizon $h$ and execution horizon $k$ of the diffusion policy. Together, the prediction and execution horizons determine the number of temporally overlapping action components (i.e., between $a_{t+k:t+h-1|t}$ and $a_{t+k:t+h-1|t+k}$) that are statistically compared by STAC, while the execution horizon governs how far apart in time the action distributions $\bar{\pi}_t$ and $\tilde{\pi}_{t+k}$ are generated.

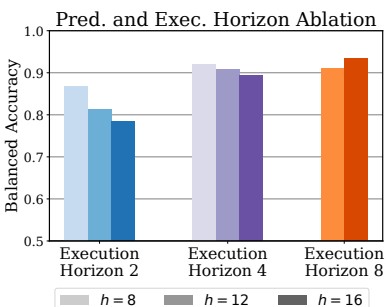

Pred. and Exec. Horizon Ablation

The result is shown in Fig. 8. We find that STAC (MMD) performs comparatively across execution horizons of $k=4$ and $k=8$, but performs best with the standard diffusion policy settings of $k=8$ and $h=16$ (used for the main result in Fig. 5). The detector's performance drops when using the smallest execution horizon of $k=2$. We attribute this to the relatively small amount of environment change that occurs within two execution steps, which causes $\bar{\pi}_t$ and $\tilde{\pi}_{t+k}$ to be similarly distributed and leads to overly conservative statistical distances. This is reflected in our results, where the detectors attain $> 95\%$ TNRs across various execution horizons, but using $k=2$ leads to a significant drop in TPR to $61\%$, while $k=4$ and $k=8$ attain TPRs of $78\%$ and $95\%$, respectively.

Figure 8: Performance variation of STAC subject to different policy prediction and execution horizons in **PushT**.

To further ablate the choice of policy prediction horizon, we conduct a similar study on the simulated **Close Box** and real-world **Push Chair** domains. The result is shown in Table 4, where we find that STAC is quite robust to the choice of prediction horizon, while the best result is achieved by using the standard setting of $h=16$.

| | Domain | Pred. Horizon ($h$) | TPR ↑ | TNR ↑ | Accuracy ↑ |
|---|---|---|---|---|---|
| **Sim.** | Close Box | 8 | 0.92 | 0.94 | 0.93 |
| | Close Box | 12 | 0.88 | 1.00 | 0.93 |
| | Close Box | 16 | 0.96 | 1.00 | 0.98 |
| **Real** | Push Chair | 8 | 1.00 | 0.80 | 0.90 |
| | Push Chair | 12 | 0.80 | 0.80 | 0.80 |
| | Push Chair | 16 | 0.80 | 0.90 | 0.85 |

Table 4: STAC ablation on policy prediction horizon $h$.

Overall, STAC's performance may vary with the policy's execution horizon, but is relatively stable across choices of the policy's prediction horizon. The fact that we calibrate STAC and deploy it with the same prediction horizon may normalize the influence of this parameter at test time.

**How does STAC's performance vary with the choice of statistical distance function?**

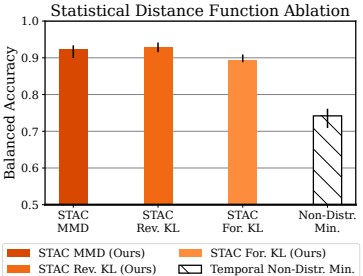

Statistical Distance Function Ablation

Fig. 9 ablates STAC's performance across various choices of statistical distance functions in the **PushT** domain. Here, we find that STAC performs comparably across common choices like maximum mean discrepancy (MMD) with RBF kernels and KL-divergence via kernel density estimation (details in §A.1). This finding is corroborated in Table 5, where all variants of STAC attain a detection accuracy of over 90% in the **Close Box** domain. In contrast, we observe a large performance drop when using a non-statistical distance function ("Temporal Non-Distr. Min"; §B.3.1) to measure temporal action consistency. This performance drop can be attributed to stochastic multimodality of the generative policy, which makes it challenging to sample individual actions that are similar to those at preceding timesteps

Figure 9: Performance variation of STAC based on the choice of statistical distance function in **PushT**.

during policy rollout. Because the non-statistical distance function is more sensitive than statistical distance functions to stochasticity in action sampling, we see an increased occurrence of false alarms.

## C.2 Extended Results: VLM Runtime Monitor

To supplement the analysis provided in §6, we herein focus on the performance of our VLM runtime monitor and its complementary role to STAC for the detection of erratic and task progression failures.

**Erratic Failure Analysis**    Table 5 presents the extended results of our experiments on the **Close Box** domain, where we aim to detect erratic policy failures that result from OOD scaling of the box. We run several evaluations of our VLM runtime monitor, varying the choice of VLM (GPT-4o, Claude 3.5 Sonnet, Gemini 1.5 Pro [79]) and prompt template (Video QA, Image QA). Since erratic failures in this domain are assigned to STAC—which detects 99% of them—we would like the VLM to avoid raising false alarms so as to keep the overall FPR of Sentinel low when the two detectors are combined.

| Category 1: Erratic Failures | Close Box: In-Distribution (Policy Success Rate: 91%) | | | Close Box: Out-of-Distribution (Policy Success Rate: 41%) | | | Close Box: Combined (Policy Success Rate: 67%) | | |
|---|---|---|---|---|---|---|---|---|---|
| Failure Detector | TPR ↑ | TNR ↑ | Det. Time (s) ↓ | TPR ↑ | TNR ↑ | Det. Time (s) ↓ | TPR ↑ | TNR ↑ | Accuracy ↑ |
| **Diffusion** Temporal Non-Distr. Min. | 1.00 | 0.97 | 5.00 | 1.00 | 0.27 | 12.35 | 1.00 | 0.77 | 0.85 |
| Diffusion Recon. [75] | 0.33 | 0.95 | 13.60 | 0.40 | 1.00 | 17.08 | 0.37 | 0.96 | 0.76 |
| Temporal Diffusion Recon. | 1.00 | 0.96 | 8.47 | 0.92 | 1.00 | 15.75 | 0.92 | 0.97 | 0.95 |
| DDPM Loss (Eq. 2) | 1.00 | 0.90 | 8.27 | 1.00 | 0.94 | 14.54 | 1.00 | 0.91 | 0.94 |
| Temporal DDPM Loss | 1.00 | 0.95 | 7.53 | 1.00 | 0.37 | 13.66 | 1.00 | 0.79 | 0.86 |
| Diffusion Output Variance | 0.33 | 0.94 | 14.00 | 0.28 | 1.00 | 17.27 | 0.26 | 0.96 | 0.72 |
| **Embed.** Policy Encoder | 0.25 | 0.98 | 16.27 | 1.00 | 0.00 | 1.59 | 0.94 | 0.70 | 0.78 |
| CLIP Pretrained | 1.00 | 0.95 | 15.73 | 1.00 | 0.00 | 8.20 | 1.00 | 0.68 | 0.79 |
| ResNet Pretrained | 1.00 | 0.95 | 17.87 | 1.00 | 0.00 | 15.51 | 1.00 | 0.68 | 0.79 |
| **STAC** STAC For. KL | 1.00 | 0.90 | 6.60 | 0.99 | 0.85 | 14.04 | 0.99 | 0.89 | 0.92 |
| STAC Rev. KL | 1.00 | 0.95 | 7.60 | 0.93 | 0.97 | 15.12 | 0.93 | 0.96 | 0.95 |
| **STAC MMD*** | 1.00 | 0.94 | 7.20 | 0.99 | 0.93 | 14.72 | 0.99 | 0.94 | 0.96 |
| **VLM** GPT-4o Image QA | 1.00 | 0.00 | 23.20 | 1.00 | 0.00 | 23.20 | 1.00 | 0.00 | 0.29 |
| **GPT-4o Video QA*** | 1.00 | 0.89 | 21.20 | 0.69 | 0.95 | 21.02 | 0.77 | 0.91 | 0.87 |
| Gemini 1.5 Pro Image QA | 1.00 | 0.00 | 21.20 | 1.00 | 0.00 | 23.20 | 1.00 | 0.00 | 0.29 |
| Gemini 1.5 Pro Video QA | 1.00 | 0.57 | 17.20 | 1.00 | 0.50 | 20.20 | 1.00 | 0.55 | 0.68 |
| Claude 3.5 Sonnet Image QA | 1.00 | 0.06 | 23.20 | 0.69 | 0.10 | 23.20 | 0.77 | 0.07 | 0.27 |
| Claude 3.5 Sonnet Video QA | 0.83 | 0.31 | 23.20 | 0.44 | 0.40 | 23.20 | 0.55 | 0.35 | 0.40 |
| **Sentinel (STAC MMD* + GPT-4o Video QA*)** | 1.00 | 0.86 | 5.47 | 1.00 | 0.90 | 14.25 | 1.00 | 0.87 | 0.91 |

Table 5: **Extended results on detecting erratic failures in the Close Box domain.** Our temporal consistency detector, STAC, detects 99% of erratic failures exhibited by diffusion policies. VLMs raise many false alarms when prompted with just a single image (Image QA), whereas performing Video QA (Ours) leads to a stark increase in TNR across all models. Performance also varies across the choice of VLM; GPT-4o is the most reliable in this domain, in contrast to other VLMs that struggle to accurately characterize the robot's task progress in the video. Overall, Sentinel detects 100% of failures, while combining STAC and the VLM increases false alarms to 13%.

We first discuss the poor performance of the Image QA baseline. When prompted with just a single image, we find that all VLMs struggle to distinguish policy successes from failures and thereby do not exceed a TNR of 7%. Without observing the initial state (i.e., the box with its lids open) and the actions of the robot, the VLM is unable to identify the lids of the box and whether they have been closed. Thus, once the task's time limit is exceeded, the VLM simply declares failure. We iterated on several prompts that included detailed questions in attempt to coerce the VLM to reason about the location of the box and its lids, but this yielded negligible changes in performance. As a result, all the outputs of the Image QA baseline resemble the following example of a false positive:

```
Analysis: The current observation shows the manipulator's arms positioned near
    the white box, with the grippers open and not grasping the lids. The two
    smaller white side lids and the bigger white back lid of the box are not
    visible, suggesting they are not yet folded. The elapsed time is 30 out of 30
     seconds, which means the robot has reached the time limit for completing
    the task. Given that the lids are not folded and the task is not completed,
    the robot is clearly failing the task.

Overall assessment: failure
```

Prompting the VLMs in a Video QA setup (§A.2) results in a strict increase in TNR across all models. Although, depending on the domain, we clearly observe that some VLMs show better visual reasoning performance than others. On the **Close Box** domain, GPT-4o generates relatively coherent descriptions of the videos, whereas Claude 3.5 Sonnet and Gemini 1.5 Pro often e.g., confuse closed lids for open lids or fail to recognize that the robot has made any significant progress. While this performance discrepancy is difficult to explain, two potential reasons are: a) the **Close Box** task can require reasoning over long videos (i.e., 20-30 image frames) and, while the VLMs' large context windows are accommodating,

the VLMs may still be susceptible to recency bias [88]; b) the images rendered in this domain might be better represented in the training data of one VLM (GPT-4o, in this case) compared to others.

**Task Progression Failure Analysis**  In the context of VLM runtime monitoring, task progression failures differ from erratic failures in that they are more visually apparent and hence simpler for the VLM to interpret. For example, the policy takes more obviously incorrect actions, e.g., clearly misplacing the cover in the **Cover Object** domain (see Fig. 7), but it does so in a temporally consistent manner that goes unnoticed by STAC. Therefore, under task progression failures, we require the VLM to attain both a high TPR and TNR, whereas we are mainly concerned with TNR under erratic failures. The extended results of our task progression failure experiments are shown in Table 6 for the **Cover Object** domain and Table 7 for the **Close Box** domain, the two of which are aggregated in Fig. 6.

| Category 2: Task Progression Failures | Cover Object: In-Distribution (Policy Success Rate: 98%) | | | Cover Object: Out-of-Distribution (Policy Success Rate: 3%) | | | Cover Object: Combined (Policy Success Rate: 56%) | | |
|---|---|---|---|---|---|---|---|---|---|
| Failure Detector | TPR↑ | TNR↑ | Det. Time (s)↓ | TPR↑ | TNR↑ | Det. Time (s)↓ | TPR↑ | TNR↑ | Accuracy↑ |
| **Diff.** Temporal Non-Distr. Min. | 1.00 | 0.93 | 2.40 | 0.06 | 1.00 | 7.60 | 0.09 | 0.93 | 0.56 |
| Diffusion Output Variance | 0.00 | 0.77 | - | 0.00 | 1.00 | - | 0.00 | 0.77 | 0.44 |
| **STAC** STAC For. KL | 1.00 | 0.95 | 2.40 | 0.03 | 1.00 | 6.40 | 0.06 | 0.95 | 0.56 |
| STAC Rev. KL | 1.00 | 0.93 | 2.40 | 0.03 | 1.00 | 6.40 | 0.06 | 0.93 | 0.55 |
| **STAC MMD*** | 1.00 | 0.93 | 2.40 | 0.09 | 1.00 | 7.73 | 0.12 | 0.93 | 0.58 |
| **VLM** GPT-4o Image QA | 1.00 | 0.07 | 12.00 | 1.00 | 0.00 | 11.03 | 1.00 | 0.07 | 0.47 |
| GPT-4o Video QA | 1.00 | 0.05 | 5.60 | 1.00 | 0.00 | 10.06 | 1.00 | 0.05 | 0.46 |
| Gemini 1.5 Pro Image QA | 1.00 | 0.05 | 12.00 | 0.91 | 0.00 | 11.15 | 0.91 | 0.05 | 0.42 |
| Gemini 1.5 Pro Video QA | 1.00 | 0.00 | 5.60 | 1.00 | 0.00 | 7.54 | 1.00 | 0.00 | 0.44 |
| Claude 3.5 Sonnet Image QA | 1.00 | 0.81 | 12.00 | 0.70 | 1.00 | 12.00 | 0.71 | 0.82 | 0.77 |
| Claude 3.5 Sonnet Video QA | 1.00 | 0.84 | 12.00 | 0.79 | 1.00 | 12.00 | 0.79 | 0.84 | 0.82 |
| Claude 3.5 Sonnet Video QA + Success Video | 1.00 | 0.77 | 12.00 | 0.94 | 1.00 | 11.59 | 0.94 | 0.77 | 0.85 |
| Claude 3.5 Sonnet Video QA + Goal Images | 1.00 | 0.93 | 5.60 | 0.76 | 1.00 | 11.74 | 0.76 | 0.93 | 0.86 |
| **Claude 3.5 Sonnet Prompt Ensemble* (see §A.2.1)** | 1.00 | 0.93 | 12.00 | 0.85 | 1.00 | 12.00 | 0.85 | 0.93 | 0.90 |
| Sentinel (STAC MMD* + Claude 3.5 Sonnet Video QA) | 1.00 | 0.79 | 2.40 | 0.82 | 1.00 | 8.68 | 0.82 | 0.80 | 0.81 |
| **Sentinel (STAC MMD* + Claude 3.5 Sonnet Prompt Ensemble*)** | 1.00 | 0.88 | 2.40 | 0.88 | 1.00 | 8.69 | 0.88 | 0.89 | 0.88 |

Table 6: **Extended results on detecting task progression failures in the Cover Object domain.** STAC only detects 12% of task progression failures (i.e., when the policy fails in a temporally consistent manner), which highlights the need for VLM runtime monitoring. Claude 3.5 Sonnet exhibits the most reliable detection performance in this domain, however, we use a prompt ensembling technique (details in §A.2.1) to reduce the number of false positives. Overall, Sentinel detects 88% of failures whilst raising 11% false alarms.

Among the VLMs considered, we find that Claude 3.5 Sonnet exhibits the best performance in the **Cover Object** domain (Table 6), achieving a 79% TPR and an 84% TNR when prompted in a Video QA setup. Qualitative analysis of the responses from GPT-4o and Gemini 1.5 Pro reveals that they misinterpret the videos and thus raise an excessive number of false alarms. As expected, STAC achieves an appreciable 93% TNR, but only detects 12% of task progression failures. The combination of STAC and Claude 3.5 Sonnet Video QA performs amicably (82% TPR, 80% TNR) but can be improved in terms of reducing the number of false positives, the majority of which are raised by the VLM. Here, we show that the prompt ensembling strategy discussed in §A.2.1 strengthens the reliability of our VLM runtime monitor ("Claude 3.5 Sonnet Prompt Ensemble"), increasing the TNR to 93%. Finally, the combination of this improved VLM runtime monitor with STAC results in a version of Sentinel that detects 88% of failures whilst raising a more acceptable number of false alarms (11%).

The results in Table 7 reaffirm the following **key takeaways**: a) image-based VLM reasoning is insufficient for understanding the robot's task progress, thus resulting in low TNRs; b) the VLMs' performances vary across domains, with GPT-4o and Claude 3.5 Sonnet performing the best in the **Close Box** and **Cover Object** domains, respectively; c) STAC and the VLM runtime monitor play complementary roles toward a performant overall failure detector across domains. We expect the performance of our VLM runtime monitor (and thus Sentinel) to improve with the future release of more capable VLMs [89], which may also eliminate the discrepancies among VLMs noted above.

**Discussion: Why combine failure detectors by taking the union of their predictions?**

Our full failure detector, Sentinel, combines STAC and the VLM by taking the union of their predictions (i.e., the "Logical OR" in Fig. 3), which, in the worst case, compounds their false positive rates by applying the union bound. However, doing so follows from several design considerations:

| Category 2: Task Progression Failures | Close Box: In-Distribution (Policy Success Rate: 85%) | | | Close Box: Out-of-Distribution (Policy Success Rate: 0%) | | | Close Box: Combined (Policy Success Rate: 40%) | | |
|---|---|---|---|---|---|---|---|---|---|
| Failure Detector | TPR↑ | TNR↑ | Det. Time (s)↓ | TPR↑ | TNR↑ | Det. Time (s)↓ | TPR↑ | TNR↑ | Accuracy↑ |
| *Diff.* Temporal Non-Distr. Min. | 1.00 | 0.97 | 4.67 | 0.67 | - | 7.46 | 0.71 | 0.97 | 0.82 |
| Diffusion Output Variance | 0.00 | 1.00 | - | 0.28 | - | 11.57 | 0.25 | 1.00 | 0.55 |
| *STAC* STAC For. KL | 1.00 | 0.97 | 5.07 | 0.61 | - | 8.14 | 0.65 | 0.97 | 0.78 |
| STAC Rev. KL | 1.00 | 0.97 | 6.13 | 0.61 | - | 10.11 | 0.65 | 0.97 | 0.78 |
| **STAC MMD*** | 1.00 | 0.97 | 5.47 | 0.61 | - | 9.06 | 0.65 | 0.97 | 0.78 |
| *VLM* GPT-4o Image QA | 1.00 | 0.00 | 23.20 | 1.00 | - | 22.68 | 1.00 | 0.00 | 0.60 |
| **GPT-4o Video QA*** | 1.00 | 0.89 | 21.20 | 0.87 | - | 22.00 | 0.88 | 0.89 | 0.89 |
| Gemini 1.5 Pro Image QA | 1.00 | 0.00 | 21.20 | 0.96 | - | 23.20 | 0.96 | 0.00 | 0.57 |
| Gemini 1.5 Pro Video QA | 1.00 | 0.57 | 17.20 | 0.98 | - | 15.47 | 0.98 | 0.57 | 0.82 |
| Claude 3.5 Sonnet Image QA | 1.00 | 0.06 | 23.20 | 0.78 | - | 23.20 | 0.81 | 0.06 | 0.51 |
| Claude 3.5 Sonnet Video QA | 0.83 | 0.31 | 23.20 | 0.80 | - | 23.20 | 0.81 | 0.31 | 0.61 |
| **Sentinel (STAC MMD* + GPT-4o Video QA*)** | 1.00 | 0.86 | 5.47 | 0.96 | - | 12.20 | 0.96 | 0.86 | 0.92 |

Table 7: **Extended results on detecting task progression failures in the Close Box domain.** Here, STAC detects considerably more task progression failures than in the **Cover Object** domain, yet 35% of failures are left undetected. As in Table 5, we find that video-based reasoning is necessary for VLMs to attain high TNRs, with GPT-4o showing the best performance. Overall, Sentinel detects 96% of failures whilst raising 14% false alarms.

- **Importance weighting:** We explicitly define one failure category as the complement of the other and assign a specialized detector to each because it is extremely difficult to design a single detector that captures highly heterogeneous failure modes. Thus, by using the "OR" operation, we are placing *equal importance* on the two proposed failure categories. We note that our failure detector's primary purpose is to detect unseen failures at deployment time: i.e., we do not assume any data of robot failures to calibrate the detector, which may be necessary to tune importance weights for different detectors. Furthermore, importance weights that are optimal on one dataset may perform poorly on failure modes not represented in that data.

- **Performance interpretability:** Using a simple scheme to combine detectors makes it easy to interpret a) the runtime behavior of the combined detector and b) which individual detectors are contributing to performance and when. For example, using the logical "OR" implies that Sentinel's overall FPR remains acceptably low if both STAC and the VLM runtime monitor have low FPRs. STAC achieves a provably low FPR (§D), while strategies exist to reduce the VLM's FPR through e.g., prompt ensembling (§A.2.1) or conformal calibration [90]. Thus, we can expect a low FPR when the detectors are combined. We can similarly interpret Sentinel's TPR performance, as exemplified in our experimental analysis. Ease of performance interpretation may not hold true with more sophisticated schemes for combining detectors.

- **Runtime constraints:** In practice, STAC (fast) and the VLM runtime monitor (slow) come with different inference-time latencies and may need to run at distinct timescales. Thus, our logical "OR" combination only applies at overlapping timesteps and otherwise allows each failure detector to flag independently, i.e., without synchronizing their detection rates. This flexibility is crucial, as more sophisticated combination schemes could encounter issues if e.g., unexpected network latencies result in delayed responses from a cloud-hosted VLM.

Nevertheless, we note that more sophisticated combination schemes might offer advantages, such as improved detection performance or scalability when integrating additional detectors. Exploring such schemes that align with the above design considerations represents a valuable direction for future work.

# D  Derivations

To validate our design choices, we show that STAC's score function and calibration procedure in §4.1 provably result in a low FPR. To do so, we apply recently popularized tools from conformal prediction because they are sample efficient and distribution free, meaning that they do not require distributional assumptions on the trajectory rollouts. Our guarantee is a direct application of the standard results in split conformal prediction [66], but to ensure the self-containedness of this manuscript, we first briefly reintroduce the core concepts in conformal prediction (taken from [66]) using the notation in our paper.

**Background on Conformal Inference**   In its most basic form, conformal prediction aims to construct a prediction set $\mathcal{C}$ that will contain the true value of a new test point $X_{\text{test}}$ with a user defined probability of at least $1-\delta$ [66]. To do so, a conformal algorithm requires 1) a sequence of calibration samples $\{X^i\}_{i=1}^M$ with all samples $X^1, ..., X^M, X_{\text{test}}$ i.i.d. and 2) a conformity score function $\eta(X) \in \mathbb{R}$. Intuitively, conformal methods use $\{\eta(X^i)\}_{i=1}^M$ to identify how likely $\eta(X_{\text{test}})$ is to lie within the range of a $1-\delta$ fraction of the calibration samples (i.e., how well $X_{\text{test}}$ conforms to the calibration data). We emphasize that this approach ensures that we construct a valid prediction set $\mathcal{C}$, regardless of the choice of conformity score and without knowing any properties of the data generating distribution:

**Theorem 1** (Adapted from Thm. D.1 in [66]).  *Let $\mathcal{D}_{\text{calib}} = \{X^1, ..., X^M\}$ be a calibration dataset and let $X_{\text{test}}$ be a test sample. Suppose that the samples in $\mathcal{D}_{\text{calib}}$ and $X_{\text{test}}$ are independent and identically distributed (i.i.d.). Then, defining*

$$\gamma := \inf\left\{\xi \in \mathbb{R} : \frac{|\{i : \eta(X^i) \le \xi\}|}{M} \ge \frac{\lceil (M+1)(1-\delta) \rceil}{M}\right\}$$

*as the $\frac{\lceil (M+1)(1-\delta) \rceil}{M}$ empirical quantile of the calibration data ensures that*

$$\mathbb{P}\big(\eta(X_{\text{test}}) \le \gamma\big) \ge 1-\delta.$$

*Here, $\lceil \cdot \rceil$ denotes the ceiling function.*

**Conformal guarantee of STAC**   The base split conformal procedure outlined by Theorem 1 requires that the samples used for calibration and test are i.i.d. This is not the case for states and actions observed sequentially within a trajectory, complicating the analysis of applying STAC at each timestep within a trajectory. Thus, to resolve this issue and provide a guarantee when we sequentially apply STAC on the correlated state-action pairs within a trajectory, we calibrate the detector using the consistency scores generated across full trajectories in §4.1. This allows us to rigorously bound the FPR using Theorem 1.

**Proposition 2** (STAC has low FPR).  *Let $\mathcal{D}_\tau = \{\tau^i\}_{i=1}^M \overset{\text{iid}}{\sim} P_\tau$ be the validation dataset of successful trajectories, each consisting of $H_i \le H$ timesteps and drawn i.i.d. from the closed-loop nominal distribution $P_\tau$. For notational simplicity, assume that any trajectory has a length divisible by $k$ (i.e., $H_i \bmod k = 0$). Moreover, let $\eta_t$ be defined as the STAC temporal consistency score at some timestep $0 \le t \le H$ in Eq. 1 and set $\gamma$ equal to the empirical $\frac{\lceil (M+1)(1-\delta) \rceil}{M}$ quantile of the terminal STAC scores $\{\eta_{H_i}^i\}_{i=1}^M$ of the trajectories in $\mathcal{D}_\tau$. Then, the* false positive rate—*that is, the probability that we raise a false alarm at any point during a new successful test trajectory $\tau \sim P_\tau$ of length $H' \le H$—is at most $\delta$:*

$$\text{FPR} := \mathbb{P}_{P_\tau}\big(\exists\, 0 \le t \le H' \text{ s.t. } \eta_t > \gamma\big) \le \delta. \tag{5}$$

*Proof.*   Let $H' \le H$ be the length of the test trajectory $\tau$. If there is no distribution shift, i.e., when the test trajectory $\tau$ is i.i.d. with respect to $\mathcal{D}_\tau \overset{\text{iid}}{\sim} P_\tau$, it holds that $\eta_{H'}$ and $\{\eta_{H_i}^i\}_{i=1}^M$ are i.i.d. Therefore, by Theorem 1, we have that

$$\mathbb{P}_{P_\tau}\big(\eta_{H'} > \gamma\big) \le \delta.$$

Moreover, since we define $\eta_t = \sum_{i=0}^{j-1} \hat{D}(\bar{\pi}_{ik}, \tilde{\pi}_{(i+1)k})$ for $t = jk$ in Eq. 1 and since $\hat{D}(\cdot, \cdot) \ge 0$ because it is a statistical distance, it follows that $\eta_t$ is increasing. That is, $\eta_0 \le \eta_k \le \eta_{2k} \le \cdots \le \eta_{H'}$. Therefore, if $\eta_t$ crosses the threshold $\gamma$ at any time, it also holds that $\eta_{H'} > \gamma$. This immediately implies the proposition, as we then have that

$$\mathbb{P}_{P_\tau}\big(\exists\, 0 \le t \le H' \text{ s.t. } \eta_t > \gamma\big) = \mathbb{P}_{P_\tau}\big(\eta_{H'} > \gamma\big) \le \delta.$$

$\square$

We conclude this section with three remarks:

1. We only bound the FPR, which ensures that our algorithm does not raise a false alarm with high probability, so that any warnings likely correspond to an OOD scenario. We do so because a

system that frequently raises false alarms is impractical to use. Our calibration approach does not guarantee the detection of failures, nor does it guarantee that we do not issue false alarms on OOD successes, as this is not possible without any distributional assumptions on the OOD scenarios or without using failure data for calibration [91]. Instead, we empirically find that our temporal consistency score performs amicably at detecting failures in our experiments.

2. Proposition 2 only certifies that the FPR of STAC is low. We make no claims on combined performance of STAC and the VLM, as VLM represents a black-box classifier. Future work could investigate methodologies to jointly calibrate an ensemble of failure detectors.

3. Conformal guarantees, like those in Theorem 1 and Proposition 2, are *marginal* with respect to the calibration data. That is, they may not hold exactly when given a particular calibration dataset, but if we were to sample thousands of calibration datasets, the guarantees would hold on average. Thus, as expected, STAC does not exactly satisfy Eq. 5 in our results, as compute budgets restricted our experiments to repetitions on a limited number of random seeds.

