# OpenReview forum: "Unpacking Failure Modes of Generative Policies: Runtime Monitoring of Consistency and Progress"
_robot-learning.org/CoRL/2024/Conference — CoRL 2024_

### Official Review · Reviewer_wii5 · 2024-07-21
**A runtime monitoring framework that detects failures samples from generative imitation policy**

**Originality:** 3
**Technical Quality:** 3
**Clarity Of Presentation:** 4
**Potential Impact:** 3
**Recommendation:** 3
**Confidence:** 3

**Review:**

Strength:

The paper addresses an important problem that often times imitation policies tend to drift and cause out-of-distribution failures. The ability to categorize failures and detect them early on is important and useful. The paper presentation is clear and figures are illustrative of the high-level ideas.

Weakness:

The paper needs to motivate better why monitoring generative policies require new tools compared to existing failure detection methods used for other policies. In particular, there is a lack of justification why the failure cases for generative policies can be exhaustively represented by two categories: lack of consistency and progress. While the paper use high true positive rate, high true negative rate, and detection time to measure the success of using STAC or VLM to detect the above two issues, it would be better to come up with metrics to evaluate how well STAC detects a lack of consistency and how well VLM detects a lack of progress separately. Lastly, the paper might need more than three experiments to demonstrate the universal usefulness of their runtime detection pipeline. Especially real-world experiments should be considered.

**Quality Of The Limitations Section:**

3

**Questions For Rebuttal:**

1. Can you show more tasks (more than the three tasks considered right now) and especially the failure cases of each task in videos to help readers understand the two category of failures in action?
2. Can you show in video how STAC catches inconsistent action prediction due to mode oscillation? I am aware you have a video of it. But neither could I visually identify the events that cause the detector to report policy failure, nor could I see the multimodal oscillatory behavior in the video.
3. Will action prediction horizon impact STAC?
4. Can you show videos of the remaining failure cases that the combination of STAC and VLM still cannot catch?

**Robotics Focus:**

3

**Summary Of Paper:**

This paper first divides generative policy failure cases into two categories: temporal inconsistency due to OOD sampling and lack of progress despite consistent actions. Secondly the paper addresses these two categories of failures with a runtime STAC metric and a VLM. The paper shows the combination of both monitoring mechanism detects policy failures the best.

**Summary Of Recommendation:**

Good paper with some weaknesses mentioned above. Addressing these concerns will position the paper better for acceptance.

---

### Official Review · Reviewer_emvf · 2024-07-29
**Good but confusing read**

**Originality:** 3
**Technical Quality:** 3
**Clarity Of Presentation:** 2
**Potential Impact:** 3
**Recommendation:** 3
**Confidence:** 2

**Review:**

The paper addresses a significant challenge in deploying robot policies, particularly in scenarios that may be out-of-distribution. Categorizing potential failures based on time sensitivity is a sound and logical approach. The use of Vision-Language Models (VLM) for failure detection aligns with current practices of leveraging foundation models.

However, while the theoretical foundation of the proposed approach is reasonable, the paper suffers from a lack of clarity, particularly in sections 3 and 4, which makes it difficult to read. There are also minor typing errors scattered throughout the main text (e.g., "Model-based methods are not not directly applicable" (line 79), "timesteps of the MPD" (line 112), etc.). Additionally, the work seems to reference generative policies (DP) heavily but it lacks sufficient background information about DP or generative policies in the related works section, making it challenging for readers without expertise in generative policies to comprehend. I still could not really understand why this work is necessary for tackling failure detection in generative policies in specific. Enhancing the explanation (e.g. what is $\epsilon^i$ in Eq. 1) and presentation of the concepts would help in better understanding and application of the framework.

In conclusion, the proposed framework in the paper is a valuable addition to existing work on failure detection but improved clarity and presentation are necessary.

Strengths:
1. Sound approach using a combination of STAC and VLM for identifying time-sensitive policy instability and long-term task progression.
2. Use of foundation model for long-term classification (success/failure), aligning with recent approaches of leveraging LLMs for works beyond NLP.

Weakness:
1. Lack of clarity and poor presentation, making reading difficult and confusing at times.

**Quality Of The Limitations Section:**

3

**Questions For Rebuttal:**

1. In section 3 on the paragraph regarding failure detection, the authors state their goal is "to construct a failure detector $f_\phi(\tau_t) \rightarrow \\{\text{ok, failure}\\}$ that at each timestep $t$, can provide a classification as to whether the policy will fail if it continues executing for the remaining $H - t$ timesteps of the MDP. Here, $\phi$ denotes the parameter of the failure detector". The use of the term "parameter" is misleading in the context of learning since $f$ is not a neural network. Instead, the approach relies of the STAC, which is a statistical measure of difference in the policy probabilities, and the VLM, which does not appear to be fined-tuned for this task. It took quite a while to realize that $\phi$ is the temporal consistency score (if I did not misunderstand). Perhaps, it will be better to clarify the failure detection module with suitable wordings. Additionally, while the stated goal is failure prediction (i.e., determining if the policy will fail if it continues), the presented approach is a reactive detection mechanism, with the STAC assessing the stability of the policy and the VLM determining it efficiency up to the current timestep.

2. How effective is the VLM in detecting policy failures? Unlike a cited reference that fine-tunes VLMs with task-related trajectories, the proposed approach does not fine-tune the VLM. What was the reason for this decision? According to Table 1, the True Positive Rate (TPR) of GPT-4o Video is on the lower end (in comparison to GPT-4o Image), while the True Negative Rate (TNR) is high. Does this imply that the VLM often fails to raise a warning when the policy fails, or that it generally tends not to raise any warning at all? Do the authors believe that better accuracy could be achieved through fine-tuning?

3. The proposed approach is a naive combination of two types of detectors using a logical OR. Do the authors have any thoughts on future works regarding enhanced combination that can be more effective or efficient?

**Robotics Focus:**

3

**Summary Of Paper:**

This paper introduces a framework for detecting and monitoring failures in robot policies, specifically generative policies like Diffusion Policies (DP). It classifies failures into two types: time-sensitive erratic failures and less time-sensitive task progression failures. To detect erratic failures, the framework recommends comparing the policy's action distributions across different time steps using the proposed Statistical measure of Temporal Action Consistency (STAC). For monitoring task progression and evaluating success, it suggests utilizing foundation models, particularly Vision-Language Models (VLM), to assess the policy's effectiveness in achieving its objectives.

**Summary Of Recommendation:**

The presented approach can be beneficial for improving failure detection in robotic environments, but I believe great amount of work needs to be done to improve clarity and presentation.

---

### Official Review · Reviewer_nTL6 · 2024-08-05
**Unpacking Failure Modes of Generative Policies: Runtime Monitoring of Consistency and Progress**

**Originality:** 3
**Technical Quality:** 3
**Clarity Of Presentation:** 3
**Potential Impact:** 3
**Recommendation:** 3
**Confidence:** 3

**Review:**

The paper proposes a failure detection scheme that concurrently assesses temporal action consistency and task progress. The former is determined by the novel STAC model, while the latter is evaluated using a proposed sequence image-based VLM model.
In the problem setup, policy failure is defined as the event when a trajectory yielding a lower reward than expected, and failure detection is formulated as a binary classification task {ok, failure}.

some comments are listed below:

- The reviewer finds a discrepancy between the definition of failure in the problem setup and the proposed scheme. The paper lacks analytical or experimental evidence demonstrating that the proposed architecture (action consistency + task progress) aligns with the notion of policy fault, characterized by lower-than-expected reward, as defined in the problem setup.

- The reviewer finds the approach of concurrently checking task progress and temporal action consistency to be intuitively sound for robust failure detection. Moreover, the proposed STAC and sequence image-based VLM models appear to be technically sound, each showing promising results when evaluated independently.

- In the overall architecture (Figure 2), the final failure decision is determined solely by an "OR" logic gate, combining the results of temporal action consistency and task progress assessments. However, in the reviewer’s understanding, STAC can cause significant false alarm because it merely checks for plan consistency, not plan effectiveness. In this point of view, the “OR” logic seems to be not sufficient because the false alarm of one can cause entire malfunction (e.g., increased FPR).

- The technical description of the VLM-based task progress check model is lacking in detail.

- In addition to the summarized statistical results, it would be helpful to visually illustrate the trajectory, decision-making process, and failure detection behavior over time for several specific experimental cases. This would provide a clearer understanding of how the proposed method operates in practice.

**Quality Of The Limitations Section:**

2

**Questions For Rebuttal:**

1. Misalignment between Failure Definition and Proposed Scheme:
  - A description of link between the proposed architecture (STAC + VLM) and the reward-based definition of failure.

2. Potential False Alarms from detectors and the Limitations of the "OR" Logic:
 - Provide a more robust justification for why "OR" logic are adopted, despite the risk of false alarms.
 - or, explore alternative decision-making mechanisms, such as weighted combinations or more sophisticated logic gates, to mitigate this risk.

3. Insufficient Technical Description of the VLM-based Task Progress Check Model:
  - more detailed and comprehensive description of the VLM-based task progress check model will be helpful

4. Lack of Case Study Illustrations:
  - In addition to the summarized statistical results, it would be helpful to visually illustrate the trajectory, decision-making process, and failure detection behavior over time for several specific experimental cases.

**Robotics Focus:**

3

**Summary Of Paper:**

The paper proposes a failure detection scheme that concurrently assesses temporal action consistency and task progress. The former is determined by the novel STAC model, while the latter is evaluated using a proposed sequence image-based VLM model.

**Summary Of Recommendation:**

While the individual modules comprising the architecture are technically sound and the results of individual module appear promising, there is room for improvement in the proposed overall architecture. I recommend a weak accept.

---

### Author Rebuttal · Authors · 2024-08-11

We would like to thank the reviewers once more for their thorough reviews and invaluable feedback. We are especially grateful for the acknowledgement of our manuscript's technical soundness and significance toward addressing a critical challenge in the deployment of learning-based robot policies. In the attached file, we include our revised manuscript with suggested changes highlighted in blue for your convenience.

# Main changes
1. Provided more background information on generative policies and motivation for new failure detection methodology in the Introduction and Related Work.
2. Revised the Problem Setup to improve clarity and highlight connections to our experimental setup.
3. Added our new real-world hardware experiments to the Experiments, Results, and Appendix B sections.
4. Refined the limitations in accordance with reviewer comments

---

### Decision · Program_Chairs · 2024-09-04

**Decision:**

Accept

**Comment:**

Strengths:
- The paper addresses a critical challenge in depolyment of robot policies.
- Theoretical foundation of the work seems to be relatively sound.

Weaknesses:
- The clarity of presentation should be improved.
- The categoriztion of the presented binary failure modes is not convincingingly descirbed.
- Real-world evaluation would be beneficial.